# Fine-grained Abnormality Prompt Learning for Zero-shot Anomaly Detection

## Abstract

Current zero-shot anomaly detection (ZSAD) methods show remarkable success in prompting large pre-trained vision-language models to detect anomalies in a target dataset without using any dataset-specific training or demonstration. However, these methods are often focused on crafting/learning prompts that capture only coarse-grained semantics of abnormality, *e.g.*, high-level semantics like 'damaged', 'imperfect', or 'defective' objects. They therefore have limited capability in recognizing diverse abnormality details that deviate from these general abnormal patterns in various ways. To address this limitation, we propose FAPrompt, a novel framework designed to learn Fine-grained Abnormality Prompts for more accurate ZSAD. To this end, we introduce a novel *compound abnormality prompting* module in FAPrompt to learn a set of complementary, decomposed abnormality prompts, where each abnormality prompt is formed by a compound of shared normal tokens and a few learnable abnormal tokens. On the other hand, the fine-grained abnormality patterns can be very different from one dataset to another. To enhance their cross-dataset generalization, we further introduce a *data-dependent abnormality prior* module that learns to derive abnormality features from each query/test image as a sample-wise abnormality prior to ground the abnormality prompts in a given target dataset. Comprehensive experiments conducted across 19 real-world datasets, covering both industrial defects and medical anomalies, demonstrate that FAPrompt substantially outperforms state-of-the-art methods by at least 3%-5% AUC/AP in both image- and pixel-level ZSAD tasks.

## 1 Introduction

Anomaly Detection (AD) is a critical task in computer vision, aiming to identify instances that deviate significantly from the majority of data. It has a wide range of real-world applications, *e.g.*, industrial inspection and medical imaging analysis (Pang et al., 2021; Cao et al., 2024). Traditional AD methods focus on learning specialized detectors with large training samples. Consequently, these methods often rely on application-specific, carefully curated datasets to train a detection model, making them inapplicable for application scenarios where such data access is not possible due to data privacy issue, or where the test data significantly differs from the training set due to substantial distribution shifts arising from new deployment environments or other natural variations in datasets. Zero-shot AD (ZSAD), which aims at learning generalist models for detecting anomalies in a target dataset without using any dataset-specific training or demonstration, has been recently emerging as a promising approach to address this limitation of traditional AD approaches.

In recent years, large pre-trained vision-language models (VLMs) such as CLIP (Radford et al., 2021) have demonstrated impressive zero/few-shot recognition capabilities across a broad range of vision tasks, including the ZSAD task (Chen et al., 2023b; Jeong et al., 2023; Deng et al., 2023; Zhou et al., 2024). To leverage VLMs for AD, the methods craft/learn text prompts to extract the textual semantic of normal/abnormal from VLMs for matching visual anomalies. These methods, such as WinCLIP (Jeong et al., 2023) and AnoVL (Deng et al., 2023), attempt to capture a range of abnormality semantics for better ZSAD by including a wide variety of pre-defined state-aware tokens (*e.g.*, using 'damaged', 'imperfect', or 'defective' to depict defects on different objects like carpet) or domain-aware tokens (*e.g.*, 'industrial', 'manufacturing', or 'surface') into the text prompts. Others (Zhou et al., 2024; 2022b;a) employ learnable text prompts to extract more general-purpose features for representing the normal/abnormal class, such

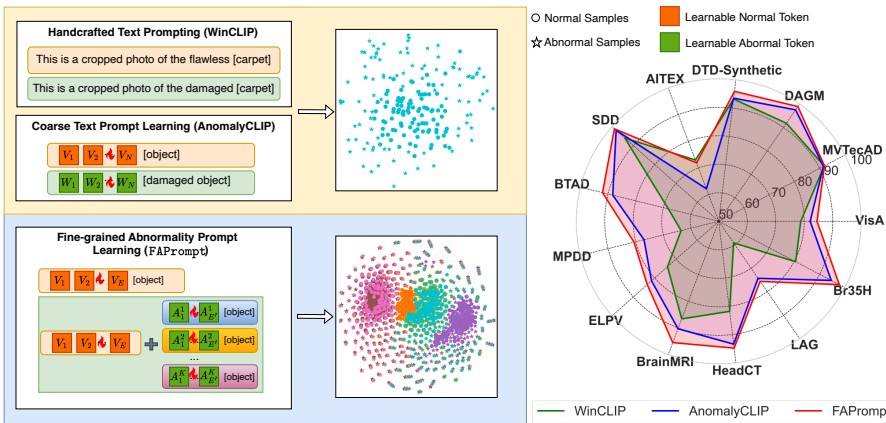

**Figure 1: Left**: `FAPrompt` vs. two related methods. **Right**: Their image-level ZSAD results in AUROC.

as AnomalyCLIP (Zhou et al., 2024). However, these methods are often focused on crafting/learning prompts that capture only coarse-grained semantics of abnormality, *e.g.*, high-level semantics like 'damaged', 'imperfect', or 'defective' objects. They therefore have limited capability in recognizing diverse abnormalities that deviate from these coarse-grained abnormal patterns in various ways, as shown in the top of Fig. 1 **Left** (see Fig. 3 in Sec. 4.1 for detailed analysis). A recent approach AnomalyGPT (Gu et al., 2023) deals with this issue by using detailed text description of abnormal objects through an additional Large Language Model (LLM), but it requires the reference samples from the target data, which is a different task from ZSAD. It also heavily relies on costly human annotations for the detailed textual descriptions.

To tackle these issues, we propose a novel framework, namely `FAPrompt`, designed to learn Fine-grained Abnormality Prompts for more accurate ZSAD. In contrast to previous prompting methods, `FAPrompt` focuses on learning the prompts that can model diverse fine-grained abnormality semantics without requiring detailed human annotations or text descriptions, as illustrated by various discriminative abnormal patterns in the bottom of Fig. 1 **Left**. To this end, in `FAPrompt` we introduce a novel *Compound Abnormality Prompting* module, namely **CAP**, to learn a set of complementary, decomposed abnormality prompts on top of a normal prompt, where each abnormality prompt is formed by a compound of the same tokens in the normal prompt and a few learnable abnormal tokens. The insight of this design is rooted from our observation that each abnormal pattern can be considered as some unexpected patterns overlaying on top of common normal patterns, *e.g.*, color stains on normal texture of carpet. Such a compound prompting strategy enables the learning of different abnormality semantics easily while maintaining abnormality prompts in good proximity to the normal prompt. This helps avoid learning trivial abnormality prompts that are too far away from the normal prompt, lacking discriminability for distinguishing normal and abnormal samples.

On the other hand, the fine-grained abnormality patterns can be very different from one dataset to another. Thus, to achieve better cross-dataset generalization, the learned fine-grained abnormality prompts should be adaptive to any target testing datasets. We therefore further introduce a *Data-dependent Abnormality Prior* module, namely **DAP**, to enhance the cross-dataset generalizability of the abnormal tokens in CAP. It learns to derive abnormality features from each query/test image as a sample-wise abnormality prior to dynamically adapt the abnormality prompts in CAP to a given target dataset. This interaction between CAP and DAP enables the learning of abnormality prompts that have fine-grained semantics and are adaptive to different testing datasets, enabling better ZSAD across a wide range of image AD datasets, as shown in Fig. 1 **Right**.

Accordingly, we make the following main contributions.

- We propose a novel ZSAD framework `FAPrompt`. Unlike existing methods that capture coarse-grained semantics of abnormality only, `FAPrompt` offers an effective approach for learning adaptive fine-grained abnormality semantics without any reliance on detailed human annotation/text description of the diverse anomaly categories.

- To achieve this, we first introduce a novel Compound Abnormality Prompting module (CAP) in `FAPrompt`. It learns a small set of complementary, decomposed abnormality prompts on top of the normal prompt via a compounding normal-abnormal token design and an orthogonal constraint among the abnormality prompts.

- We further introduce a Data-dependent Abnormality Prior module (DAP). It learns to select the most relevant abnormal features from anomaly images while refraining from normal images for adapting the fine-grained abnormalities learned in CAP to a given target dataset.

- Comprehensive experiments on 19 diverse real-world industrial and medical image AD datasets show that `FAPrompt` significantly outperforms state-of-the-art ZSAD models by at least 3%-5% AUC/AP in both image- and pixel-level detection tasks.

## 2 RELATED WORK

### 2.1 CONVENTIONAL ANOMALY DETECTION

There have been different types of AD approaches introduced over the years. In particular, one-class classification methods (Tax & Duin, 2004; Yi & Yoon, 2020; Bergman & Hoshen, 2020; Chen et al., 2022; Ruff et al., 2020) aim to compactly describe normal data using support vectors. Reconstruction-based methods (Akcay et al., 2019; Schlegl et al., 2019; Zavrtanik et al., 2021b; Yan et al., 2021; Zaheer et al., 2020; Zavrtanik et al., 2021a; Park et al., 2020; Hou et al., 2021; Xiang et al., 2023; Liu et al., 2023; Yao et al., 2023b;a) train models to reconstruct normal images, with anomalies identified through higher reconstruction errors. Distance-based methods (Pang et al., 2018; Defard et al., 2021; Cohen & Hoshen, 2020; Roth et al., 2022) detect anomalies by measuring the distance between the test image and normal images. Knowledge distillation methods (Deng & Li, 2022; Bergmann et al., 2020; Salehi et al., 2021; Wang et al., 2021; Cao et al., 2023; Tien et al., 2023; Zhang et al., 2023) focus on distilling normal patterns from pre-trained models and detecting anomalies by comparing discrepancies between the distilled and original features. However, these methods often rely on application-specific datasets to train the detection model, limiting their applicability in real-world scenarios where data access is restricted due to privacy concerns, proprietary restrictions, or resource constraints. Also, these approaches tend to struggle when there is a significant difference between the distribution of the training and test data.

### 2.2 ZERO-SHOT ANOMALY DETECTION

ZSAD has been made possible due to the development of large pre-trained foundation models, such as vision-language models (VLMs). CLIP (Radford et al., 2021) has been widely used as a VLM to enable ZSAD on visual data (Jeong et al., 2023; Zhou et al., 2024; Deng et al., 2023; Chen et al., 2023a). CLIP-AC adapts CLIP for ZSAD by using text prompts designed for the ImageNet dataset as in (Radford et al., 2021). By using manually defined textual prompts specifically designed for industrial AD dataset, WinCLIP (Jeong et al., 2023) achieves better ZSAD performance compared to CLIP-AC, but it often does not generalize well to non-defect AD datasets. APRIL-GAN (Chen et al., 2023a) adapts CLIP to ZSAD through tuning some additional linear layers with annotated auxiliary AD data. AnoVL (Deng et al., 2023) introduces domain-aware textual prompts and test time adaptation in CLIP to enhance the ZSAD performance. AnomalyCLIP (Zhou et al., 2024) employs learnable, object-agnostic textual prompts to extract more general-purpose text features for the normal and abnormal classes. All these methods are focused on crafting/learning prompts that capture only coarse-grained semantics of abnormality, failing to detect anomalies that exhibit different patterns from these coarse abnormal patterns. There are a number of other studies leveraging CLIP for AD, but they are designed for empowering few-shot (Gu et al., 2023; Zhu & Pang, 2024) or conventional AD task (Joo et al., 2023; Wu et al., 2024a;c;b).

## 3 METHODOLOGY

### 3.1 PRELIMINARIES

**Problem Statement.** Let $\mathcal{D}_{train} = \{X_{train}, Y_{train}\}$ denote an auxiliary training dataset consisting of both normal and anomalous samples, where $X_{train} = \{x_i\}_{i=1}^N$ is a set of $N$ images and $Y_{train} = \{y_i, \mathbf{G}_i\}_{i=1}^N$ contains the corresponding ground truth labels and pixel-level anomaly masks. Each

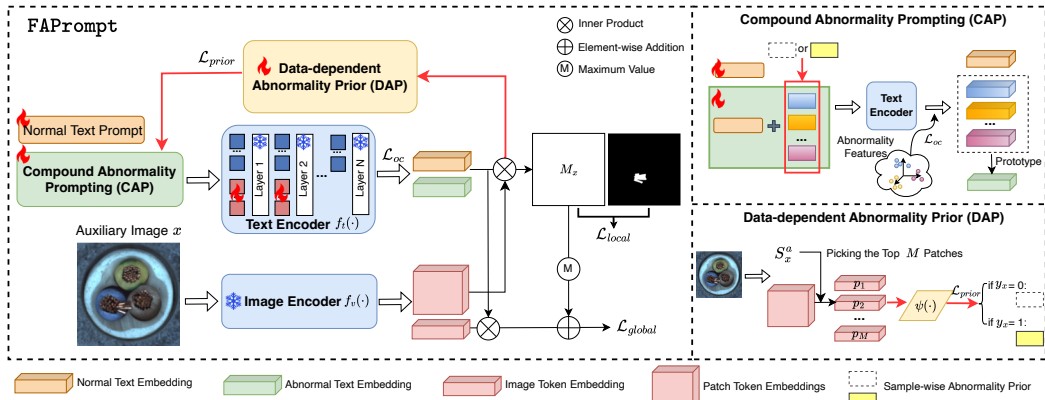

**Figure 2:** Overview of FAPrompt. It consists of two novel modules, including the Compound Abnormality Prompting (CAP) module and the Data-dependent Abnormality Prior (DAP) module detailed in the top-right and bottom-right corners respectively. CAP is devised to learn fine-grained abnormality semantics without relying on detailed human annotations or text descriptions, while DAP is designed to adaptively select the most abnormal features from each query/test image as a sample-wise abnormality prior to enhance the cross-dataset generalizability of the abnormality prompts in CAP.

image $x_i$ is labeled by $y_i$, where $y_i = 0$ indicates a normal image and $y_i = 1$ signifies an anomalous one. The anomaly mask $\mathbf{G}_i$ provides pixel-level annotation of $x_i$. During the testing phase, we are presented with a collection of target datasets, $\mathcal{T} = \{\mathcal{D}_{test}^1, \mathcal{D}_{test}^2, \cdots, \mathcal{D}_{test}^t\}$, where each $\mathcal{D}_{test}^j = \{X_{test}^j, Y_{test}^j\}$ is a test set from a target application dataset that have different normal and abnormal samples from those in the training data $\mathcal{D}_{train}$. The goal of ZSAD is to develop models on the auxiliary dataset $\mathcal{D}_{train}$, with the ability to generalize to detect anomalies in different test sets in $\mathcal{T}$. Particularly, given an input RGB image $x \in \mathbb{R}^{h \times w \times 3}$ from $\mathcal{D}_{train}$, with $h$ and $w$ respectively representing the height and width of $x$, a ZSAD model is required to output both an image-level anomaly score $s_x \in \mathbb{R}$ and a pixel-level anomaly map $\mathcal{M}_x \in \mathbb{R}^{h \times w}$. The image-level anomaly score $s_x$ provides a global assessment of whether the image is anomalous, while the pixel-level anomaly map $\mathcal{M}_x$ indicates the likelihood of each pixel being anomalous. Both $s_x$ and the values in $\mathcal{M}_x$ lie in [0, 1], where a larger value indicates a higher probability of being abnormal.

**VLM Backbone.** To enable accurate ZSAD, large pre-trained VLMs are typically required. Following existing approaches (Chen et al., 2023b; Deng et al., 2023; Jeong et al., 2023; Zhou et al., 2024), the pre-trained CLIP (Radford et al., 2021) is used in our study, which comprises a visual encoder $f_v(\cdot)$ and a text encoder $f_t(\cdot)$, where the visual and text representations are well-aligned through pre-training on web-scaled text-image pairs.

## 3.2 OVERVIEW OF FAPROMPT

In this work, we propose a ZSAD framework FAPrompt to learn adaptive fine-grained abnormality semantics without any reliance on detailed human annotations or text descriptions. Fig 2 illustrates the overall framework of FAPrompt that consists of two novel modules, including Compound Abnormality Prompting module (**CAP**) and Data-dependent Abnormality Prior module (**DAP**). To be more specific, the proposed CAP module is devised to specify the design of fine-grained abnormality prompts. The key characteristic of CAP is to obtain the abnormality prompts via a compound prompting method, where we have one normal prompt and multiple abnormality prompts are added on top of it. These normal and abnormal text prompts are then processed by the CLIP's text encoder $f_t(\cdot)$ to generate the corresponding normal and abnormal text embeddings, respectively. For a given image $x$, FAPrompt extracts both an image token embedding $f_v(x)$ and a set of patch token embeddings $\mathbf{F}_v \in \mathbb{R}^{l \times d}$, with $l$ and $d$ respectively representing the length and dimensionality of $\mathbf{F}_v$. The prompts are then learned using $\mathcal{D}_{train}$ based on the similarity between the image and text embeddings, where the fine-grained abnormality prompts are aggregated into an abnormality prompt prototype before its use in similarity calculation. Further, the DAP module is introduced to improve cross-dataset generalization capability of the fine-grained abnormality prompts. DAP derives the most relevant abnormality features based on the given query/test image $x$, serving as a sample-wise abnormality prior to dynamically adapt the abnormality prompts in CAP to the characteristics of a given target dataset. During training, the original parameters of CLIP remain frozen, and only

the attached learnable tokens in the text encoder layers, along with the normal and fine-grained abnormality prompts, are optimized. Below we present these modules in detail.

### 3.3 Compound Abnormality Prompt Learning

**Learning Fine-grained Abnormalities via Compound Normal and Abnormal Tokens.** Previous approaches that rely on coarse-grained learnable text prompts fail to capture the fine-grained abnormality semantics for detecting diverse anomalies across different datasets. To address this, we propose the novel CAP module. The core insight is that abnormal samples typically exhibit different magnitude of deviation from their normal counterparts while still belonging to the same class. CAP models this by learning a set of complementary, decomposed abnormality prompts built on a shared normal prompt. Following previous work (Zhou et al., 2024), a set of learnable normal tokens and the fixed token 'object' are concatenated to define the normal text prompt $\mathcal{P}^n$. For the abnormality prompt, CAP aims to learn a small set of prompts of complementary semantics, denoted as $\mathcal{P}^a = \{\mathcal{P}^{a_1}, \mathcal{P}^{a_2}, ... \mathcal{P}^{a_K}\}$, where each $\mathcal{P}^{a_i}$ is formed by a compound of the same tokens in the normal prompt $\mathcal{P}^n$ and a few learnable abnormal tokens. Formally, the normal and abnormality prompts can be defined as:

$$
\begin{aligned}
\mathcal{P}^n &= [V_1][V_2]...[V_E][object], \\
\mathcal{P}^a &= \{\mathcal{P}^{a_1}, \mathcal{P}^{a_2}, ... \mathcal{P}^{a_K}\}, \\
with \; \mathcal{P}^{a_i} &= [V_1][V_2]...[V_E][A_1^i][A_2^i]...[A_{E'}^i][object],
\end{aligned}
\tag{1}
$$

where $\{V_1, V_2, ...V_E\}$ and $\{A_1^i, A_2^i, ...A_{E'}^i\}_{i=1}^K$ are learnable normal and abnormal tokens, respectively. This compound prompting strategy enables the learning of different abnormality semantics easily while maintaining abnormality prompts in good proximity to the normal prompt, supporting the learning of non-trivial, semantically-meaningful abnormality prompts.

**Learning Complementary Abnormality Prompts.** To capture complementary fine-grained abnormalities and reduce redundant information captured by the abnormality prompts, it is essential to maximize the diversity among the fine-grained abnormalities. A straightforward approach would be to train distinct abnormal prompts on separate, annotated subsets with samples from different anomalous types. However, this would require extensive human annotations. To address this issue, we propose to add an orthogonal constraint loss $\mathcal{L}_{oc}$ into the abnormality prompts in CAP as a alternative method to encourage this diversity. Formally, the objective for this can be formulated as:

$$
\mathcal{L}_{oc} = \sum_{i,j \in K; i \neq j} abs\left(\frac{f_t(\mathcal{P}^{a_i}) \cdot f_t(\mathcal{P}^{a_j})}{||f_t(\mathcal{P}^{a_i})|| \times ||f_t(\mathcal{P}^{a_j})||}\right),
\tag{2}
$$

where the text encoder $f_t(\cdot)$ is used to extract the embeddings of the abnormality prompts, $[\cdot]$ denotes inner product, $abs(\cdot)$ returns the absolute value, and $|| \cdot ||$ indicates the norm of vectors.

To provide more representative embedding for the fine-grained abnormalities, we compute the prototype of the multiple abnormality prompt embeddings as the final fine-grained abnormality embedding via $\mathbf{F}_a = \frac{1}{|\mathcal{P}^a|} \sum_{\mathcal{P}^{a_i} \in \mathcal{P}^a} f_t(\mathcal{P}^{a_i})$. The normal text prompt embedding is $\mathbf{F}_n = f_t(\mathcal{P}^n)$.

### 3.4 Learning to Select Data-dependent Abnormality Prior

One issue in ZSAD is that the fine-grained abnormality patterns can be very different from the auxiliary dataset to test datasets. In addition to the learning of a set of complementary fine-grained abnormality prompts, it is important to ensure that the learned fine-grained abnormality patterns are generalized to target testing datasets. Inspired by the instance-conditional information design in CoCoOp (Zhou et al., 2022a), we introduce the DAP module to enhance the cross-dataset generalizability of the abnormal tokens in CAP by adaptively selecting the embeddings of the most abnormal regions to serve as a sample-wise abnormality prior for each image input. Particularly, given a query/test image $x$, DAP selects the most abnormal image patches as the abnormality prior to be fed into CAP for assisting the abnormality prompt learning. It achieves this by picking the top $M$ patches whose token embeddings are most similar to the abnormality prompt prototype $\mathbf{F}_a$:

$$
\mathbf{S}_x^a(i,j) = \frac{\exp(\mathbf{F}_v(i,j)\mathbf{F}_a^\intercal)}{\exp(\mathbf{F}_v(i,j)\mathbf{F}_n^\intercal) + \exp(\mathbf{F}_v(i,j)\mathbf{F}_a^\intercal)},
\tag{3}
$$

where $[\cdot]^{\mathsf{T}}$ denotes a transpose operation, $\mathbf{F}_v(i, j)$ is the token embedding of the patch centered at $(i, j)$ and $\mathbf{S}_x^a(i, j)$ is a patch-level anomaly score. The corresponding normal scores can be calculated via $\mathbf{S}_x^n(i, j)$ using the similarity to $\mathbf{F}_n$ in the numerator in Eq. 3.

Let $\mathbf{p}_x = \{p_1, p_2, ...p_M\}$ be the top $M$ patch embeddings of $x$, FAPrompt then adds additional learnable layers $\psi(\cdot)$, namely *abnormality prior network*, to model the sample-wise abnormality prior based on $\mathbf{p}_x$. This prior $\Omega_x = \psi(\mathbf{p}_x)$ is then incorporated as data-dependent abnormal features into the learnable abnormal tokens of the abnormality prompts in CAP to dynamically adapt the learned fine-grained abnormalities to a given target dataset, with each individual abnormality prompt refined as follows:

$$\hat{\mathcal{P}}^{a_i} = [V_1][V_2]...[V_E][A_1^i \oplus \Omega_x][A_2^i \oplus \Omega_x]...[A_{E'}^i \oplus \Omega_x][object], \quad (4)$$

where $\Omega_x$ is a vector-based prior of the same dimensionality as the abnormal tokens and $\oplus$ denotes element-wise addition. Thus, the abnormality prompt set is updated as $\hat{\mathcal{P}}^a = \{\hat{\mathcal{P}}^{a_1}, \hat{\mathcal{P}}^{a_2}, ...\hat{\mathcal{P}}^{a_K}\}$, and the abnormality prompt prototype can be accordingly refined as $\hat{\mathbf{F}}_a = \frac{1}{|\hat{\mathcal{P}}^a|} \sum_{\hat{\mathcal{P}}^{a_i} \in \hat{\mathcal{P}}^a} f_t(\hat{\mathcal{P}}^{a_i})$.

The goal of DAP is to introduce sample-wise *abnormality* information. However, there is no abnormality from the top $M$ patches of normal images, and thus, simply applying the prior $\Omega_x$ to normal images would introduce noise into the learnable abnormal tokens, damaging the learning of fine-grained abnormalities. To address this issue, we propose an abnormality prior learning loss $\mathcal{L}_{prior}$ to enforce that $\Omega_x$ is the features mapped from the most abnormal $M$ patches if $x$ is an abnormal image, while it is minimized to be a null vector if it is a normal image. Formally, $\mathcal{L}_{prior}$ can be defined as follows:

$$\mathcal{L}_{prior} = \sum_{y_x=0} \sum_{\omega \in \Omega_x} \omega_x^2, \quad (5)$$

where $\omega$ is an entry of $\Omega_x$.

### 3.5 TRAINING AND INFERENCE

**Training.** During training, FAPrompt first generates an abnormality-oriented segmentation map $\hat{\mathcal{M}}^a \in \mathbb{R}^{h \times w}$ using $\hat{S}_x^a$ whose entries are calculated via Eq. 3 with $\mathbf{F}_a$ replaced by the prior-enabled $\hat{\mathbf{F}}_a$:

$$\hat{\mathcal{M}}^a = \Phi(\hat{S}_x^a), \quad (6)$$

where $\Phi(\cdot)$ is a reshape and interpolation function that transforms the patch-level anomaly scores into a two-dimensional segmentation map. In the same way, we can generate the segmentation map $\hat{\mathcal{M}}^n = \Phi(\hat{S}_x^n)$ based on the prior-enabled normal score $\hat{S}_x^n$. Let $\mathbf{G}_x$ represent the ground-truth mask of the query image $x$, following AnomalyCLIP (Zhou et al., 2024), the learning objective in FAPrompt for optimizing pixel-level AD can then be defined as:

$$\mathcal{L}_{local} = \frac{1}{N} \sum_{x \in X_{train}} \mathcal{L}_{Focal}([\hat{\mathcal{M}}^n, \hat{\mathcal{M}}^a], \mathbf{G}_x) + \mathcal{L}_{Dice}(\hat{\mathcal{M}}^a, \mathbf{G}_x) + \mathcal{L}_{Dice}(\hat{\mathcal{M}}^n, \mathbf{I} - \mathbf{G}_x), \quad (7)$$

where $\mathbf{I}$ is a full-one matrix, $\mathcal{L}_{Focal}(\cdot)$ and $\mathcal{L}_{Dice}(\cdot)$ denote a focal loss (Lin et al., 2017) and a Dice loss (Li et al., 2019b), respectively. To ensure the accuracy of locating the top abnormal features in DAP, we apply the same learning objective to optimize the segmentation maps $\mathcal{M}^{\mathbf{n}} \in \mathbb{R}^{h \times w}$ and $\mathcal{M}^{\mathbf{a}} \in \mathbb{R}^{h \times w}$, which are derived from the normality-oriented scores $S_x^n$ and abnormality-oriented scores $S_x^a$, respectively.

For image-level supervision, FAPrompt first computes the probability of the query image $x$ being classified as abnormal based on its cosine similarity to the two prompt embeddings $\hat{\mathbf{F}}_a$ and $\mathbf{F}_n$:

$$s_a(x) = \frac{\exp(f_v(x)\hat{\mathbf{F}}_a^{\mathsf{T}})}{\exp(f_v(x)\mathbf{F}_n^{\mathsf{T}}) + \exp(f_v(x)\hat{\mathbf{F}}_a^{\mathsf{T}})}. \quad (8)$$

The final image-level anomaly score is then defined as the average of this image-level score and the maximum pixel-level anomaly score derived from the anomaly score maps:

$$s(x) = \frac{1}{2}(s_a(x) + s'_a(x)), \tag{9}$$

where $s'_a(x) = \frac{1}{2}\left(\max(S^a_x) + \max(\hat{S}^a_x)\right)$ represents the average of the maximum anomaly scores from $S^a_x$ and $\hat{S}^a_x$. Following previous methods (Zhu & Pang, 2024; Chen et al., 2023a; Zhou et al., 2024; Jeong et al., 2023), $s'_a(x)$ is treated as a complementary anomaly score to $s_a(x)$ and incorporated into Eq. 9, as $s'_a(x)$ are helpful for detecting local abnormal regions. The image-level anomaly score $s(x)$ is then optimized by minimizing the following loss on $X_{train}$:

$$\mathcal{L}_{global} = \frac{1}{N} \sum_{x \in X_{train}} \mathcal{L}_b(s(x), y_x), \tag{10}$$

where $\mathcal{L}_b$ is specified by a focal loss function due to the class imbalance in $X_{train}$. Overall, `FAPrompt` is optimized by minimizing the following combined loss, which integrates both local and global objectives, along with the two constraints from the CAP and DAP modules:

$$\mathcal{L} = \mathcal{L}_{local} + \mathcal{L}_{global} + \mathcal{L}_{prior} + \mathcal{L}_{oc}, \tag{11}$$

**Inference.** During inference, given a test image $x'$, it is fed through the visual encoder of CLIP to generate the segmentation maps $\mathcal{M}^n$, $\mathcal{M}^a$, $\hat{\mathcal{M}}^n$, and $\hat{\mathcal{M}}^a$. Then the pixel-level anomaly map $\mathcal{M}_{x'}$ is calculated by averaging over these segmentation maps as follows:

$$\mathcal{M}_{x'} = \frac{1}{4}(\mathcal{M}^a \oplus 1 \ominus \mathcal{M}^n \oplus \hat{\mathcal{M}}^a \oplus 1 \ominus \hat{\mathcal{M}}^n), \tag{12}$$

where $\ominus$ is element-wise subtraction. The image-level anomaly score $s_{x'}$ is computed using Eq. 9.

## 4 Experiments

**Datasets.** To verify the effectiveness of `FAPrompt`, we conduct extensive experiments across 19 publicly available datasets, including nine popular industrial defect inspection datasets on varying products/objects (MVTecAD (Bergmann et al., 2019), VisA (Zou et al., 2022), DAGM (Wieler & Hahn, 2007), DTD-Synthetic (Aota et al., 2023), AITEX (Silvestre-Blanes et al., 2019), SDD (Tabernik et al., 2020), BTAD (Mishra et al., 2021), MPDD (Jezek et al., 2021), and ELPV(Deitsch et al., 2019)) and ten medical anomaly detection datasets on different organs like brain, fundus, colon, skin and thyroid (BrainMRI (Salehi et al., 2021), HeadCT (Salehi et al., 2021), LAG (Li et al., 2019a), Br35H (Hamada, 2020), CVC-ColonDB (Tajbakhsh et al., 2015), CVC-ClinicDB (Bernal et al., 2015), Kvasir (Jha et al., 2020), Endo (Hicks et al., 2021), ISIC (Gutman et al., 2016), TN3K (Gong et al., 2021)) (see `Appendix A` for details about the datasets).

To assess the ZSAD performance, the models are trained on the MVTecAD dataset by default and evaluated on the test sets of other datasets without any further training or fine-tuning. We obtain the ZSAD results on MVTecAD by changing the training data to the VisA dataset.

**Competing Methods and Evaluation Metrics.** We compare our method, `FAPrompt`, with several state-of-the-art (SotA) methods, including five handcrafted text prompt-based methods – raw CLIP (Radford et al., 2021), CLIP-AC and WinCLIP (Jeong et al., 2023), APRIL-GAN (Chen et al., 2023a), and AnoVL (Deng et al., 2023) – and three learnable text prompt-based methods – CoOp (Zhou et al., 2022b), CoCoOp (Zhou et al., 2022a), and AnomalyCLIP (Zhou et al., 2024). As for evaluation metrics, we follow previous works (Jeong et al., 2023; Zhou et al., 2024) and use two popular metrics: AUROC (Area Under the Receiver Operating Characteristic) and average precision (AP) to assess the image-level AD performance; for pixel-level AD performance, we employ AUROC and Area under per region overlap (PRO) to provide a more detailed analysis.

**Table 1:** Image-level ZSAD results (AUROC, AP) on 13 AD datasets. The best and second-best results are respectively highlighted in red and blue. The results for MVTecAD, VisA, DAGM, DTD-Synthetic, BTAD, and MPDD are averaged performance across their multiple data subsets (see `Appendix` D for breakdown results).

| Data Type | Dataset | Handcrafted Text Prompts | | | | | Learnable Text Prompts | | | |
|---|---|---|---|---|---|---|---|---|---|---|
| | | CLIP | CLIP-AC | WinCLIP | APRIL-GAN | AnoVL | CoOp | CoCoOp | AnomalyCLIP | FAPrompt |
| Industrial | MVTecAD | (74.1, 87.6) | (71.5, 86.4) | (91.8, 96.5) | (86.2, 93.5) | (92.5, 96.7) | (88.8, 94.8) | (71.8, 84.9) | (91.5, 96.2) | (91.9, 95.7) |
| | VisA | (66.4, 71.4) | (65.0, 70.2) | (78.8, 81.4) | (78.0, 81.4) | (79.2, 81.7) | (62.8, 68.1) | (78.1, 82.3) | (82.1, 85.4) | (84.5, 86.8) |
| | SDD | (95.5, 87.9) | (94.7, 77.9) | (94.0, 87.2) | (97.5, 93.4) | (95.3, 91.3) | (96.8, 90.0) | (89.9, 50.4) | (98.1, 93.4) | (98.6, 95.9) |
| | BTAD | (34.5, 52.5) | (51.0, 62.1) | (68.2, 70.9) | (73.6, 68.6) | (80.3, 73.1) | (66.8, 77.4) | (48.4, 53.9) | (88.3, 87.3) | (92.0, 92.2) |
| | MPDD | (54.3, 65.4) | (56.2, 66.0) | (63.6, 69.9) | (73.0, 80.2) | (68.9, 71.9) | (55.1, 64.2) | (61.0, 69.1) | (77.0, 82.0) | (80.6, 83.3) |
| | AITEX | (71.0, 45.7) | (71.5, 46.7) | (73.0, 54.7) | (57.6, 41.3) | (72.5, 55.4) | (66.2, 39.0) | (48.6, 37.8) | (62.2, 40.4) | (71.9, 53.2) |
| | DAGM | (79.6, 59.0) | (82.5, 63.7) | (91.8, 79.5) | (94.4, 83.8) | (89.7, 76.3) | (87.5, 74.6) | (96.3, 85.5) | (97.5, 92.3) | (98.9, 95.7) |
| | DTD-Synthetic | (71.6, 85.7) | (66.8, 83.2) | (93.2, 92.6) | (86.4, 95.0) | (94.9, 97.3) | (83.1, 91.9) | (84.1, 92.9) | (93.5, 97.0) | (95.9, 98.3) |
| | ELPV | (59.2, 71.7) | (69.4, 80.2) | (74.0, 86.0) | (65.5, 79.3) | (70.6, 83.0) | (73.0, 86.5) | (78.4, 89.2) | (81.5, 91.3) | (83.5, 92.0) |
| Medical | BrainMRI | (73.9, 81.7) | (80.6, 86.4) | (86.6, 91.5) | (89.3, 90.9) | (88.7, 91.3) | (61.3, 44.9) | (78.2, 86.7) | (90.3, 92.2) | (95.5, 95.6) |
| | HeadCT | (56.5, 58.4) | (60.0, 60.7) | (81.8, 80.2) | (89.1, 89.4) | (81.6, 84.2) | (78.4, 78.8) | (80.3, 73.4) | (93.4, 91.6) | (94.8, 93.5) |
| | LAG | (58.7, 76.5) | (58.2, 76.9) | (59.2, 74.8) | (73.6, 84.8) | (65.1, 78.0) | (69.6, 82.9) | (72.6, 84.7) | (74.3, 84.9) | (75.6, 85.4) |
| | Br35H | (78.4, 78.8) | (82.7, 81.3) | (80.5, 82.2) | (93.1, 92.9) | (88.4, 88.9) | (86.0, 87.5) | (85.7, 89.1) | (94.6, 94.7) | (97.8, 97.5) |

**Table 2:** Pixel-level ZSAD results (AUROC, PRO) on 14 AD datasets. The best and second-best results are respectively highlighted in red and blue. Note that medical datasets in Table 1 do not have pixel-level ground truth. Thus, different medical datasets are used here. Detailed breakdown results for MVTecAD, VisA, DAGM, DTD-Synthetic, BTAD, and MPDD can be found in `Appendix` D.

| Data Type | Dataset | Handcrafted Text Prompts | | | | | Learnable Text Prompts | | | |
|---|---|---|---|---|---|---|---|---|---|---|
| | | CLIP | CLIP-AC | WinCLIP | APRIL-GAN | AnoVL | CoOp | CoCoOp | AnomalyCLIP | FAPrompt |
| Industrial | MVTecAD | (38.4, 11.3) | (38.2, 11.6) | (85.1, 64.6) | (87.6, 44.0) | (89.8, 76.2) | (33.3, 6.6) | (86.7, 79.6) | (91.1, 81.4) | (90.6, 83.3) |
| | VisA | (46.6, 14.8) | (47.8, 17.2) | (79.6, 56.8) | (94.2, 86.8) | (89.9, 71.2) | (24.1, 3.8) | (93.6, 86.7) | (95.5, 87.0) | (95.9, 87.5) |
| | SDD | (28.4, 5.1) | (33.5, 7.6) | (95.9, 78.4) | (93.0, 84.6) | (97.9, 82.6) | (91.8, 81.7) | (93.7, 85.0) | (98.1, 95.2) | (98.3, 93.6) |
| | BTAD | (30.6, 4.4) | (32.8, 8.3) | (72.7, 27.3) | (60.8, 25.0) | (93.2, 62.8) | (28.6, 3.8) | (86.1, 72.0) | (94.2, 74.8) | (95.6, 75.2) |
| | MPDD | (62.1, 33.0) | (58.7, 29.1) | (76.4, 48.9) | (94.1, 83.2) | (84.0, 61.0) | (15.4, 2.3) | (95.2, 84.2) | (96.5, 87.0) | (96.5, 87.9) |
| | AITEX | (53.2, 15.3) | (47.3, 11.8) | (62.5, 41.5) | (78.2, 68.8) | (59.2, 49.1) | (67.7, 54.9) | (52.1, 56.9) | (83.0, 66.5) | (82.0, 62.6) |
| | DAGM | (28.2, 2.9) | (32.7, 4.8) | (87.6, 65.7) | (82.4, 66.2) | (92.0, 78.8) | (17.5, 2.1) | (95.6, 91.0) | (98.3, 95.4) | (98.3, 95.4) |
| | DTD-Synthetic | (33.9, 12.5) | (23.7, 5.5) | (83.9, 57.8) | (95.3, 86.9) | (97.5, 90.4) | (55.8, 36.0) | (93.7, 83.7) | (97.9, 92.3) | (98.3, 93.1) |
| Medical | CVC-ColonDB | (49.5, 15.8) | (49.5, 11.5) | (70.3, 32.5) | (78.4, 64.6) | (77.9, 49.8) | (40.5, 2.6) | (79.1, 69.7) | (81.9, 71.3) | (84.6, 74.7) |
| | CVC-ClinicDB | (47.5, 18.9) | (48.5, 12.6) | (51.2, 13.8) | (80.5, 60.7) | (82.1, 55.0) | (34.8, 2.4) | (83.4, 68.8) | (82.9, 67.8) | (84.7, 70.1) |
| | Kvasir | (44.6, 17.7) | (45.0, 16.8) | (69.7, 24.5) | (75.0, 36.2) | (72.5, 28.2) | (44.1, 3.5) | (79.1, 38.6) | (78.9, 45.6) | (81.2, 47.8) |
| | Endo | (45.2, 15.9) | (46.6, 12.6) | (68.2, 28.3) | (81.9, 54.9) | (80.5, 47.7) | (40.6, 3.9) | (83.1, 59.0) | (84.1, 63.6) | (86.4, 67.2) |
| | ISIC | (33.1, 5.8) | (36.0, 7.7) | (83.3, 55.1) | (89.4, 77.2) | (90.6, 79.8) | (51.7, 15.9) | (81.9, 68.9) | (89.7, 78.4) | (90.9, 81.2) |
| | TN3K | (42.3, 7.3) | (35.6, 5.2) | (70.7, 39.8) | (73.6, 37.8) | (80.9, 50.5) | (34.0, 9.5) | (72.4, 41.0) | (81.5, 50.4) | (84.5, 54.1) |

**Implementation Details.** Following previous approaches (Zhou et al., 2024; Chen et al., 2023a), we implement `FAPrompt` using the same CLIP implementation, OpenCLIP (Ilharco et al., 2021), using the publicly available pre-trained `VIT-L/14@336px` backbone. The parameters of both the visual and text encoders in CLIP are kept frozen. By default, learnable token embeddings are attached to the first nine layers of the text encoder, with a token length of four for each layer. The lengths of the learnable normal and abnormal text prompts are respectively set to five and two by default. The number of fine-grained abnormality prompts and selected patch tokens in the DAP module are both set to ten. We use the Adam optimizer with an initial learning rate of 1e-3. Further implementation details for `FAPrompt` and the competing methods are provided in `Appendix` B.

## 4.1 MAIN RESULTS

**Image-level ZSAD Performance.** Table 1 presents the image-level ZSAD results of `FAPrompt`, compared to eight SotA methods across 13 AD datasets, including nine industrial defect AD datasets and four medical AD datasets. The results show that `FAPrompt` significantly outperforms the SotA models across almost all datasets. On average, compared to the best competing methods, it achieves up to 3.7% AUROC and 4.9% AP on industrial AD datasets and 5.2% AUROC and 3.4% AP on medical AD datasets. In particular, the weak performance of CLIP and CLIP-AC can be attributed to its over-simplified text prompt design. By utilizing more carefully designed handcrafted prompts, WinCLIP achieves better results than CLIP and CLIP-AC while preserving the training-free nature. APRIL-GAN and AnoVL improve over WinCLIP by using additional learnable layers and/or domain-aware tokens within the textual prompts. However, they heavily rely on sensitive handcrafted textual prompts and capture mainly coarse-grained semantics of abnormality, leading to poor performance when faced with anomalies that does not fit well to the pre-defined text descriptions, *e.g.*, BTAD, MPDD, BrainMRI, HeadCT, and Br35H.

As for text prompt learning methods, CoOp and CoCoOp are designed for general vision tasks, *i.e.*, discriminating different objects, so they have weak capability in capturing the differences between normality and abnormality on the same object. AnomalyCLIP significantly improves performance by learning object-agnostic textual prompts for AD, demonstrating strong generalization capabilities across diverse datasets. However, AnomalyCLIP overlooks fine-grained abnormality details. `FAPrompt` overcomes this limitation via its two novel modules, CAP and DAP.

**Pixel-level ZSAD Performance.** We also compare the pixel-level ZSAD results of our `FAPrompt` with SotA methods across 14 AD datasets in Table 2. Similar observations can be derived as the image-level results. In particular, CLIP and CLIP-AC are the weakest among the handcrafted text prompt-based methods, primarily due to inappropriate text prompt designs. With better prompt engineering (and adaptation to AD in some cases), WinCLIP, APRIL-GAN, and AnoVL demonstrate better performance. For the learnable text prompt approaches, CoOp shows poor performance due to overfitting on the adaptation dataset, while CoCoOp mitigates this limitation by introducing instance-conditional information, achieving substantial improvement over CoOp and competitive performance to AnomalyCLIP. `FAPrompt` demonstrates superior performance in identifying a wide range of pixel-level anomalies, significantly outperforming SotA models across nearly all datasets. It surpasses the best competing methods by up to 2.7% AUROC and 4.4% AP on the industrial AD datasets, and by 3.0% AUROC and 3.7% AP on the medical AD datasets. This demonstrates the effectiveness of the fine-grained abnormality prompt in `FAPrompt` that adaptively capture detailed abnormality semantics in different datasets.

**Performance of Learning Complementary Abnormalities.** To assess the complementarity of the abnormality prompts learned by `FAPrompt`, we empirically evaluate the discriminability of each abnormality prompt and its difference to the rest of

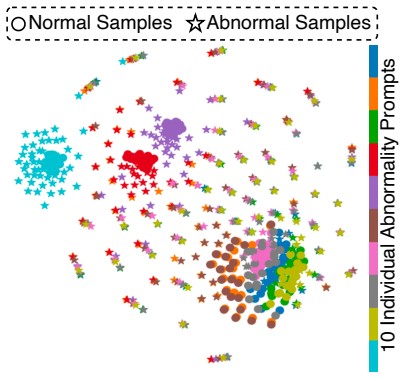

other prompts. To this end, we calculate the patch-level anomaly scores $\mathbf{S}_x^a$ of an image based on the similarity of its patch token embeddings to each individual abnormal prompt embedding (rather than the prototype of the abnormal prompt embeddings) in **CAP**, and subsequently project the anomaly scores of each sample into a two-dimensional space via t-SNE. As depicted in Fig. 3 on an exemplar dataset, two key observations can be derived: i) despite having slight overlapping, the normal and abnormal samples are distributed into a different group for each individual abnormality prompt, indicating the learning of different abnormal patterns per abnormality prompt; and ii) there is clear separation between normal and abnormal samples for the use of each abnormality prompt in anomaly scoring, indicating the good discriminability of each prompt learned in `FAPrompt`. Similar patterns can be found in more visualization and comparison with the baselines in `Appendix` C.3.

**Figure 3:** t-SNE visualization of prompt-wise anomaly scores on BTAD (01).

### 4.2 ABLATION STUDY

**Module Ablation.** Our ablation study results based on averaged performance across 18 industrial and medical datasets are shown in Table 3, where AnomalyCLIP is used as our base model (**Base**) and each of our two modules is separately added on this base model (*i.e.*, '+ **CAP**' and '+ **DAP**'). The dataset-wise performance and module ablation on single abnormality prompt can be found in `Appendix` D.2 and C.5, respectively. It can be seen that applying CAP alone results in a significant improvement in image-level ZSAD performance due to its ability in learning the fine-grained abnormality details. To assess how important the orthogonal constraint loss ($\mathcal{L}_{oc}$) is in CAP, we further evaluate the performance with $\mathcal{L}_{oc}$ removed, denoting as '+ **CAP** w\o $\mathcal{L}_{oc}$'. The results indicate that the orthogonal constraints imposed by $\mathcal{L}_{oc}$ help the CAP module work in a more effective way, justifying its effectiveness in encouraging the learning of unique and complementary fine-grained abnormal patterns in CAP.

As shown in Table 3, when DAP is applied independently, it results in substantial improvements in not only image-level performance but also pixel-level performance.

**Table 3:** Image-level (AUROC, AP) and pixel-level (AUROC, PRO) results of ablation study.

| Model | Industrial Datasets | | Medical Datasets | |
|---|---|---|---|---|
| | Image-level | Pixel-level | Image-level | Pixel-level |
| AnomalyCLIP | (85.0, 83.6) | (94.4, 84.8) | (87.7, 90.6) | (83.2, 62.9) |
| + CAP | (88.1, 87.0) | (94.6, 83.9) | (90.6, **93.1**) | (83.8, 63.8) |
| + CAP w\o $\mathcal{L}_{oc}$ | (87.2, 86.3) | (94.3, 83.5) | (90.3, 91.8) | (83.6, 63.8) |
| + DAP | (86.9, 85.2) | (94.8, 84.9) | (90.2, 92.3) | (84.6, 64.8) |
| + DAP w\o $\mathcal{L}_{prior}$ | (86.5, 85.1) | (94.7, 83.7) | (89.9, 92.3) | (84.5, 64.3) |
| AnomalyCLIP Ensemble | (85.5, 84.0) | (94.7, **85.0**) | (89.3, 91.3) | (83.2, 62.4) |
| AnomalyCLIP Ensemble* | (85.5, 82.6) | (94.6, 84.5) | (88.8, 91.0) | (83.5, 65.6) |
| `FAPrompt` | (**88.2, 87.2**) | (**95.0, 85.0**) | (**90.9**, 93.0) | (**85.4, 65.9**) |

The improvement is clearer on the medical datasets. This can be attributed to its ability of deriving data-dependent abnormality information from any target data to enhance the cross-dataset generalization of `FAPrompt`. We similarly assess the importance of the abnormality prior selection

loss ($\mathcal{L}_{prior}$) in DAP by having the variant, 'DAP w\o $\mathcal{L}_{prior}$' that removes $\mathcal{L}_{prior}$ from DAP. The results show that removing $\mathcal{L}_{prior}$ may introduce irrelevant priors from normal samples and lead to a significant drop in pixel-level performance. When all components are applied, the full model `FAPrompt` achieves its best performance. This shows that the interaction between CAP and DAP enables the learning of abnormality prompts that capture fine-grained semantics and are adaptive to different test datasets.

**`FAPrompt` vs Ensemble Methods.** To learn more abnormalities, a straightforward solution is to ensemble existing ZSAD methods. We hence conduct two ensemble strategies in AnomalyCLIP for comparison: i) to learn an ensemble of AnomalyCLIP with each learning a abnormality prompt tuned on the auxiliary dataset with a different random seed ('AnomalyCLIP Ensemble'), and ii) to learn AnomalyCLIP with an ensemble of multiple abnormality prompts with orthogonal constraint loss ('AnomalyCLIP Ensemble*').

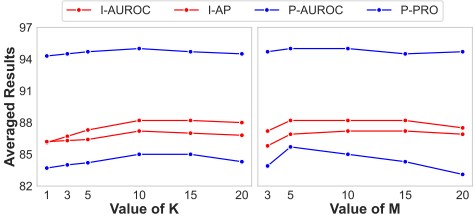

The results in Table 3 show that two simple ensemble methods can improve AnomalyCLIP to some extent, but their abnormality prompts are much less effective than `FAPrompt` as these simple strategies lead to learning of highly redundant abnormality prompts, rather than the complementary prompts learned in `FAPrompt`. This showcases the effectiveness of the abnormality prompts learned in `FAPrompt` in capturing the fine-grained abnormality details which cannot be learned in simple prompt ensemble approaches.

**Figure 4:** Averaged results on industrial datasets with varying $K$ and $M$.

**Hyperparameter Sensitivity Analysis.** We analyze the sensitivity of two key hyperparameters of `FAPrompt` on industrial datasets in terms of image-level ('I-AUROC' and 'I-AP') and pixel-level ('P-AUROC' and 'P-PRO') ZSAD performance in Fig. 4, including the number of abnormality prompts $K$ in CAP and the number of selected patch tokens $M$ in DAP (similar results can be found for medical datasets in `Appendix`. C.5).

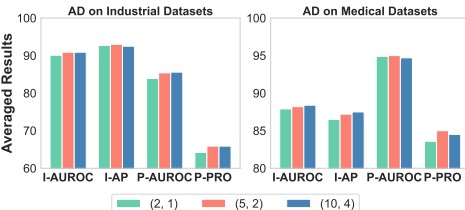

In particular, the performance gets improved with increasing $K$, typically peaking at $K = 10$. The performance may slightly declines when $K$ is chosen beyond 10. This suggests that while increasing the number of prompts helps capture a wider range of abnormalities, too large $K$ values may introduce noise or redundancy into the prompts. As for the number of selected tokens, $M$, the performance exhibits a similar pattern, with the best performance obtained at a medium value. This

**Figure 5:** Averaged results of `FAPrompt` with varying prompt sizes of $(E, E')$.

suggests that selecting too many abnormal patch candidates may introduce noise or less relevant patches into CAP, leading to the learning of less effective fine-grained anomalies. Additionally, we also evaluate the sensitivity of the length of learnable normal and abnormal tokens $\{E, E'\}$ in **CAP** module. The Image-level and pixel-level ZSAD results are shown in Fig. 5. Overall, the setting of (5, 2) works best for both industrial and medical AD, yielding strong ZSAD performance. Longer prompt lengths, such as (10, 4), can introduce more complexity without clear performance improvement, particularly in pixel-level performance. Using shorter prompt lengths, *e.g.*, the setting of (2, 1), lacks sufficient capacity to support the ZSAD task, leading to consistently weaker performance.

## 5 CONCLUSION

In this paper, we propose `FAPrompt`, a novel framework designed to enhance CLIP's performance in ZSAD by learning adaptive fine-grained abnormality semantics. `FAPrompt` introduces a Compound Abnormality Prompting (CAP) module that generates complementary abnormality prompts without relying on exhausting human annotations. Additionally, it incorporates a Data-dependent Abnormality Prior (DAP) module, which refines these prompts to improve cross-dataset generalization. The interaction between CAP and DAP enables the model to learn adaptive fine-grained abnormality semantics. Extensive experiments on 19 datasets demonstrate that `FAPrompt` significantly outperforms state-of-the-art ZSAD methods.

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

**Table 4:** Data statistics of MVTec AD and VisA.

| Dataset | Subset | Type | Original Training | Original Test | |
|---|---|---|---|---|---|
| | | | Normal | Normal | Anomalous |
| MVTec AD | Carpet | Texture | 280 | 28 | 89 |
| | Grid | Texture | 264 | 21 | 57 |
| | Leather | Texture | 245 | 32 | 92 |
| | Tile | Texture | 230 | 33 | 83 |
| | Wood | Texture | 247 | 19 | 60 |
| | Bottle | Object | 209 | 20 | 63 |
| | Capsule | Object | 219 | 23 | 109 |
| | Pill | Object | 267 | 26 | 141 |
| | Transistor | Object | 213 | 60 | 40 |
| | Zipper | Object | 240 | 32 | 119 |
| | Cable | Object | 224 | 58 | 92 |
| | Hazelnut | Object | 391 | 40 | 70 |
| | Metal_nut | Object | 220 | 22 | 93 |
| | Screw | Object | 320 | 41 | 119 |
| | Toothbrush | Object | 60 | 12 | 30 |
| VisA | candle | Object | 900 | 100 | 100 |
| | capsules | Object | 542 | 60 | 100 |
| | cashew | Object | 450 | 50 | 100 |
| | chewinggum | Object | 453 | 50 | 100 |
| | fryum | Object | 450 | 50 | 100 |
| | macaroni1 | Object | 900 | 100 | 100 |
| | macaroni2 | Object | 900 | 100 | 100 |
| | pcb1 | Object | 904 | 100 | 100 |
| | pcb2 | Object | 901 | 100 | 100 |
| | pcb3 | Object | 905 | 101 | 100 |
| | pcb4 | Object | 904 | 101 | 100 |
| | pipe_fryum | Object | 450 | 50 | 100 |

## A DATASET DETAILS

### A.1 DATA STATISTICS OF TRAINING AND TESTING

We conduct extensive experiments on 19 real-world Anomaly Detection (AD) datasets, including nine industrial defect inspection datasets (MVTecAD (Bergmann et al., 2019), VisA (Zou et al., 2022), DAGM (Wieler & Hahn, 2007), DTD-Synthetic (Aota et al., 2023), AITEX (Silvestre-Blanes et al., 2019), SDD (Tabernik et al., 2020), BTAD (Mishra et al., 2021), MPDD (Jezek et al., 2021), ELPV(Deitsch et al., 2019)) and ten medical anomaly detection datasets (BrainMRI (Salehi et al., 2021), HeadCT (Salehi et al., 2021), LAG (Li et al., 2019a), Br35H (Hamada, 2020), CVC-ColonDB (Tajbakhsh et al., 2015), CVC-ClinicDB (Bernal et al., 2015), Kvasir (Jha et al., 2020), Endo (Hicks et al., 2021), ISIC (Gutman et al., 2016), TN3K (Gong et al., 2021)).

To assess the ZSAD performance, the full dataset of MVTec AD, including both training set and test set, is used as the auxiliary training data, on which AD models are trained, and they are subsequently evaluated on the test set of the other 18 datasets without any further training. We train the model on the full dataset of VisA when evaluating the performance on MVTec AD. Table 4 provides the data statistics of MVTec AD and VisA, while Table 5 shows the test set statistics of the other 17 datasets.

## B IMPLEMENTATION DETAILS

### B.1 DETAILS OF MODEL CONFIGURATION.

Following previous works (Deng et al., 2023; Chen et al., 2023a; Zhou et al., 2024), `FAPrompt` adopts a modified version of CLIP –OpenCLIP (Ilharco et al., 2021) and its publicly available pre-trained backbone `VIT-L/14@336px`– as the VLM backbone to enhance the model's attention to local features while preserving its original structure. Following Zhou et al. (2024), we replace the original Q-K self-attention mechanism in the visual encoder with a V-V self-attention mechanism during patch feature extraction, starting from the 6th layer of the visual encoder. The parameters of both the visual and text encoders in CLIP are frozen throughout the experiments.

**Table 5:** Data statistics of the other 17 AD datasets. They are used for ZSAD inference only.

| Data type | Dataset | Modalities | \|C\| | Normal | Anomalous |
|---|---|---|---|---|---|
| **Object** | **SDD** | Photography | 1 | 286 | 54 |
| | **BTAD** | Photography | 3 | 451 | 290 |
| | **MPDD** | Photography | 6 | 176 | 282 |
| **Textual** | **AITEX** | Photography | 12 | 564 | 183 |
| | **DAGM** | Photography | 10 | 6996 | 1054 |
| | **DTD-Synthetic** | Photography | 12 | 357 | 947 |
| | **ELPV** | Electroluminescence | 2 | 377 | 715 |
| **Brain** | **BrainMRI** | Radiology (MRI) | 1 | 98 | 155 |
| | **HeadCT** | Radiology (CT) | 1 | 100 | 100 |
| | **Br35H** | Radiology (MRI) | 1 | 1500 | 1500 |
| **Fundus** | **LAG** | Fundus Photography | 1 | 786 | 1711 |
| **Colon** | **CVC-ColonDB** | Endoscopy | 1 | 0 | 380 |
| | **CVC-ClinicDB** | Endoscopy | 1 | 0 | 612 |
| | **Kvasir** | Endoscopy | 1 | 0 | 1000 |
| | **Endo** | Endoscopy | 1 | 0 | 200 |
| **Skin** | **ISIC** | Photography | 1 | 0 | 379 |
| **Thyroid** | **TN3K** | Radiology (Utralsound) | 1 | 0 | 614 |

Inspired by previous works (Jia et al., 2022; Zhou et al., 2024; Khattak et al., 2022), We use text prompt tuning to refine the original textual space of CLIP by adding additional learnable token embeddings into its text encoder. By default, the learnable token embeddings are attached to the first 9 layers of the text encoder to refine the textual space, with a token length of four for each layer. The lengths of the learnable normal prompt and abnormal tokens in CAP are set to five and two, respectively. The number of fine-grained abnormality prompts ($K$) and selected patch tokens ($M$) in DAP are both set to 10. To align with the dimension of VIT-L/14@336px, the abnormality prior network $\psi(\cdot)$ is configured with the input and output dimensions of $768 \times M$ and 768, respectively, and includes a hidden layer of size $(768 \times M)/16$ with ReLU activation.

We utilize the Adam optimizer with an initial learning rate of 1e-3 to update the model parameters. The input images are resized to 518×518 with a batch size of eight. This resizing is also applied to other baseline models for a fair comparison, while preserving their original data preprocessing methods, if applicable. The training is conducted for seven epochs across all experiments. During the inference stage, a Gaussian filter with $\sigma = 10$ is applied to smooth the anomaly score map. All experiments are conducted using PyTorch on a single GPU (NVIDIA GeForce RTX 3090).

### B.2 Implementation of Comparison Methods

To evaluate the efficiency of FAPrompt, we compare its performance against eight state-of-the-art (SotA) baselines. The results for CLIP (Ilharco et al., 2021), CLIP-AC (Ilharco et al., 2021), WinCLIP (Jeong et al., 2023), APRIL-GAN (Chen et al., 2023a), CoOp (Zhou et al., 2022b), and AnomalyCLIP (Zhou et al., 2024) are sourced from AnomalyCLIP, except the newly added datasets (SDD, AITEX, ELPV, LAG). For fair comparison, these implementations follow the setup of AnomalyCLIP. We use the official implementations of AnoVL (Deng et al., 2023) and CoCoOp (Zhou et al., 2022a). To adapt CoCoOp for ZSAD, we replace its learnable text prompt templates with normality and abnormality text prompt templates, which is consistent with the implementation of CoOp in existing ZSAD studies. All other parameters remain consistent with those specified in their original papers.

## C Additional Results

### C.1 Model Complexity of FAPrompt vs. SotA Methods

We compare the model complexity of FAPrompt with SotA methods in Table 6, evaluating the number of parameters, per-batch training time, and per-image inference time. The batch size for all approaches is set to eight for fair comparison, excluding training-free methods WinCLIP and AnoVL. While FAPrompt introduces additional trainable parameters, leading to a slightly longer training time, this minor computational overhead results in substantial performance

**Table 6:** Number of parameters, per-batch training time (ms) and per-image inference time (ms) in comparison with competing methods.

| Model | Number of Para. | Training Time | Inference Time |
|---|---|---|---|
| WinCLIP | 0 | 0 | $227.5_{\pm 0.7}$ |
| AnoVL | 0 | 0 | $171.4_{\pm 0.5}$ |
| APRIL-GAN | 3148800 | $368.7_{\pm 0.5}$ | $47.9_{\pm 0.1}$ |
| CoOp | 9216 | $643.8_{\pm 1.1}$ | $89.9_{\pm 0.7}$ |
| CoCoOp | 83760 | $737.4_{\pm 3.6}$ | $93.8_{\pm 0.7}$ |
| AnomalyCLIP | 5555200 | $914.1_{\pm 0.9}$ | $124.2_{\pm 0.9}$ |
| FAPrompt | 9612256 | $1354.1_{\pm 1.7}$ | $214.7_{\pm 0.8}$ |

improvements over competing methods. Additionally, since training is performed offline, this training computational overhead is generally negligible in real-world applications. In terms of inference time, our approach remains reasonably efficient and responsive.

## C.2 Comparison with SOTA Full-shot Methods and Prompt Tuning Methods

We conduct experiments on five of the most commonly used datasets to examine the performance gap between FAPrompt and two SotA full-shot methods, PatchCore (Roth et al., 2022) and RD4AD (Deng & Li, 2022). Note that it is not a fair comparison as PatchCore and RD4AD utilize the full training data of each testing dataset in its detection while ZSAD methods like FAPrompt does not use any of such training data. The results presented in Table 7 are only for analyzing the possible upper bound performance of ZSAD. Despite the unfair utilization of the dataset-specific training data in PatchCore and RD4AD, FAPrompt obtains rather impressive detection performance, further reducing the performance gap between ZSAD and full-shot methods.

We also compare FAPrompt with SotA prompt tuning approache TCP (Yao et al., 2024) to further verify the effectiveness of fine-grained abnormality prompt. Sine TCP is not originally designed for anomaly detection and its contextual information relies heavily on handcrafted text prompts, we adapted TCP for the ZSAD by testing two types of AD-oriented text prompts, resulting in two variants of TCP for ZSAD, **TCP_V1** and **TCP_V2**:

- **TCP_V1**, where we use a straightforward prompt design: the normal prompt is in the form of "This is a photo of [cls]." while the abnormal prompt is in the form of "This is a photo of damaged [cls]."

- **TCP_V2**, where we adopt the complete set of the prompt templates from WinCLIP.

For a fair comparison, we maintained the original model designs of TCP throughout the experiments. As shown in Table 8, both TCP variants largely underperform AnomalyCLIP and FAPrompt in the ZSAD task. This is primarily due to the fact that TCP is not designed for ZSAD and also has strong reliance on handcrafted text prompts.

In contrast, FAPrompt is specifically designed for the ZSAD task, leveraging data-dependent abnormality prior of the query images to learn complementary abnormality prompts. This adaptive approach enables FAPrompt to more effectively capture a wide variety of anomalies, resulting in promising performance in both image-level and pixel-level ZSAD tasks.

## C.3 t-SNE Visualization of Prompt-wise Anomaly Scores

To explore the complementarity of abnormality prompts in FAPrompt, we provide two-dimensional t-SNE visualization of the anomaly score map $S_x^a$ and quantitative results of 'AnomalyCLIP', prompt ensemble method 'AnomalyCLIP Ensemble*' for their comparison with FAPrompt on the three datasets. The results are shown in Fig. 6. Note that the difference between AnomalyCLIP and FAPrompt/AnomalyCLIP Ensemble* in the figure is because AnomalyCLIP learns one single abnormality prompt only while the FAPrompt/AnomalyCLIP Ensemble* learns 10 abnormality prompts.

**Table 7:** Comparison of ZSAD performance between `FAPrompt` and two SotA full-shot methods. The best and second-best results are respectively highlighted in red and blue.

| Dataset | AnomalyCLIP | FAPrompt | PatchCore | RD4AD |
|---|---|---|---|---|
| **Image-level (AUROC, AP)** | | | | |
| MVTecAD | (91.5, 96.2) | (91.9, 95.7) | ( 99.0, 99.7) | (98.7, 99.4) |
| VisA | (82.1, 85.4) | (84.5, 86.8) | (94.6, 95.9) | ( 95.3, 95.7) |
| BTAD | (88.3, 87.3) | (92.0, 92.2) | (93.2, 98.6) | ( 93.8, 96.8) |
| MPDD | (77.0, 82.0) | (80.6, 83.3) | ( 94.1, 96.3) | (91.6, 93.8) |
| DAGM | (97.5, 92.3) | ( 98.9, 95.7) | (92.7, 81.3) | (92.9, 79.1) |
| **Pixel-level (AUROC, PRO)** | | | | |
| MVTecAD | (91.1, 81.4) | (90.6, 83.3) | ( 98.1, 92.8) | (97.8, 93.6) |
| VisA | (95.5, 87.0) | (95.9, 87.5) | ( 98.5, 92.2) | (98.4, 91.2) |
| BTAD | (94.2, 74.8) | (95.6, 75.2) | (97.4, 74.4) | ( 97.5, 75.1) |
| MPDD | (96.5, 87.0) | (96.5, 87.9) | ( 98.8, 94.9) | (98.4, 95.2) |
| DAGM | (95.6, 91.0) | ( 98.3, 95.4) | (95.9, 87.9) | (96.8, 91.9) |

**Table 8:** Comparison with TCP.

| Model | Industrial | | Medical | |
|---|---|---|---|---|
| | image-level | pixel-level | image-level | pixel-level |
| AnomalyCLIP | (85.0, 83.6) | (94.4, 84.8) | (87.7, 90.6) | (83.2, 62.9) |
| TCP_V1 | (61.3, 55.9) | (87.2, 66.6) | (56.4, 61.7) | (80.2, 60.9) |
| TCP_V2 | (64.9, 59.1) | (88.5, 71.5) | (53.3, 60.3) | (76.8, 52.9) |
| Ours | (88.2, 87.2) | (95.0, 85.0) | (90.9, 93.0) | (85.4, 65.9) |

**`FAPrompt` vs. AnomalyCLIP.** It is clear that compared to AnomalyCLIP, `FAPrompt` learns a set of effective complementary abnormal patterns captured by the 10 abnormality prompts, resulting in better detection performance on datasets with complex anomaly cases.

For example, on the datasets BTAD(01) and VisA (pcb4), several anomalies, which are distributed very closely to, or overlapped with part of the normal images, are difficult to detect using single abnormality prompt in AnomalyCLIP, indicating that its single abnormality prompt is not discriminative w.r.t. these anomalies. `FAPrompt` alleviates this situation with the abnormality prompts that show visually different, discriminative power.

For datasets with simpler patterns like VisA (chewinggum), single abnormality prompt is sufficient, while having multiple abnormality prompts in `FAPrompt` do not have adverse effect. This demonstrates the performance of `FAPrompt` in achieving stable, effective detection across simple and complex datasets.

**`FAPrompt` vs. the prompt ensemble method 'AnomalyCLIP Ensemble*'.** Despite also learning multiple abnormality prompts, it is clear from the visualization that the abnormality prompts in AnomalyCLIP Ensemble* tend to be clustered closely, while that in `FAPrompt` is much more disperse, *e.g.*, two clustered patterns on BTAD(01) and one clustered pattern on VisA (pcb4) learned by AnomalyCLIP Ensemble* vs. four disperse patterns on both datasets learned by `FAPrompt`. Importantly, the more disperse abnormal patterns from `FAPrompt` provides complementary discriminative power to each other, substantiated by the enhanced AUROC/AP performance compared to AnomalyCLIP Ensemble*.

## C.4 COMPARISON WITH ALTERNATIVES TO AVERAGING STRATEGY IN CAP

Despite the simplicity, the use of the averaging operation is due to its general effectiveness in aggregating multiple patterns. This strategy is also widely used in existing ZSAD and FSAD methods, such as WinCLIP and AnoVL, to deal with diverse and complementary abnormality text information To validate its advantage over the alternatives, we conduct additional experiments to evaluate two variants of `FAPrompt`, with the results presented in Table 9:

- **$FAPrompt_{0.1}$: Selecting the most similar prompt for each detected abnormality.** In this variant of `FAPrompt`, we calculate the cosine similarity between the individual abnormality prompts and each test image to select the similarity to the most similar prompt

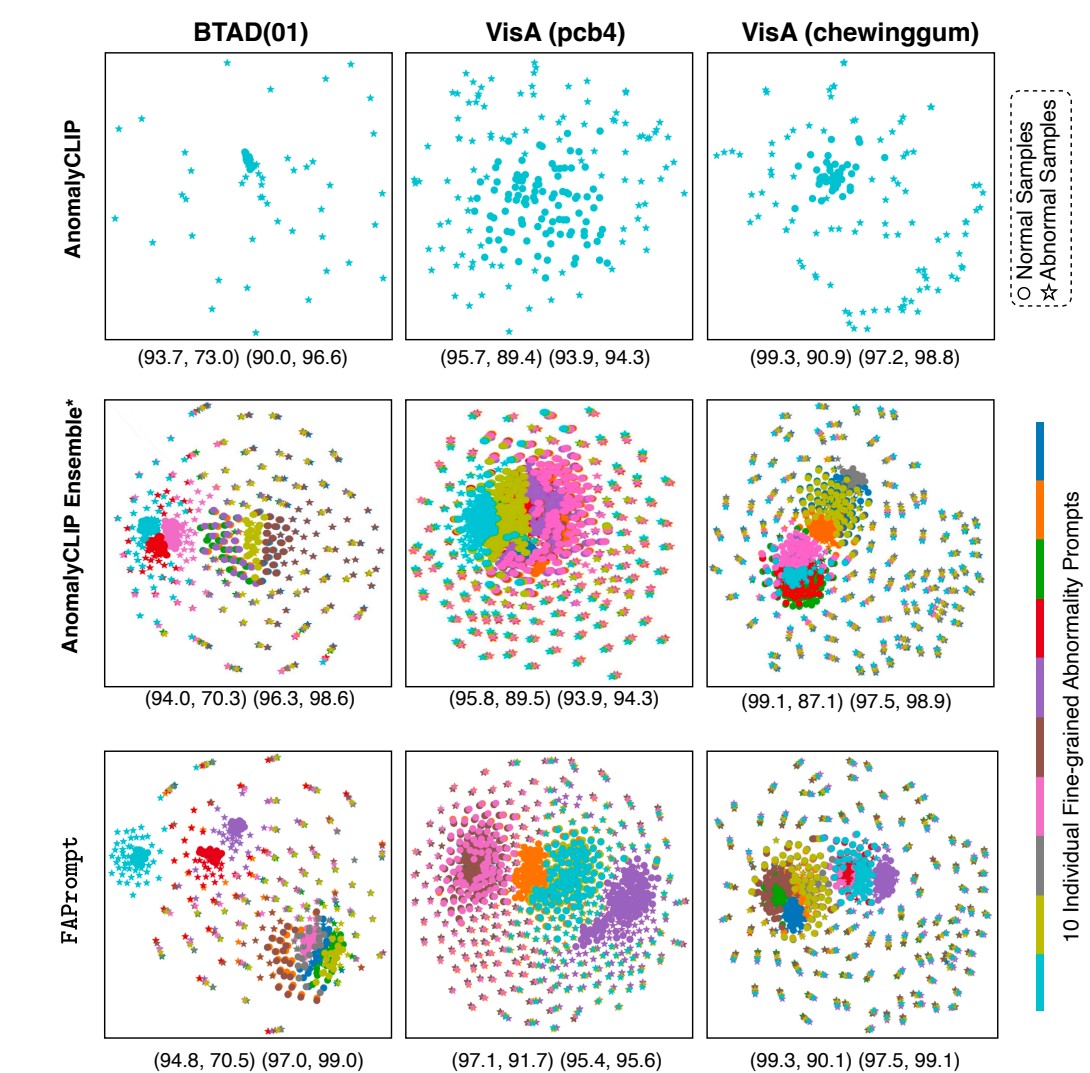

**Figure 6:** 2-D t-SNE visualizations and quantitative results (Image-level AUROC, Image-level AP) (Pixel-level AUROC, Pixel-level PRO) of `FAPrompt`, AnomalyCLIP and its ensemble method AnomalyCLIP Ensemble*.

as the anomaly score during inference. While this approach shows comparable performance on image-level ZSAD results, it can largely underperform the primary `FAPrompt` in pixel-level ZSAD. This is mainly because selecting only a single prompt can lead to the loss of complementary information from other abnormality prompts, limiting the model's ability to detect the full spectrum of abnormalities.

- **`FAPrompt`$_{0.2}$: Using weighted abnormality prompts.** In this variant, we use a prompt importance learning network to learn a set of weights for each abnormality prompt based on the selected most abnormal patch tokens of the query images. These weights are then used to combine multiple abnormality prompts into a single weighted abnormality prompt (a weighted abnormality prototype) for ZSAD. Although `FAPrompt`$_{0.2}$ outperforms `FAPrompt`$_{0.1}$ by retaining some complementary abnormality information, it does not match the performance of the simple averaging. This may be due to the greater power of the model in fitting the query images, which can lead to overfitting of the tuning auxiliary dataset in the zero-shot setting, *i.e.*, the learned weights may well reflect the significance of each prompt in the tuning dataset but not in the target datasets.

**Table 9:** Comparison with alternatives to averaging the abnormality prompts in `FAPrompt`.

| Model | Industrial Datasets | | Medical Datasets | |
|---|---|---|---|---|
| | Image-level | Pixel-level | Image-level | Pixel-level |
| **AnomalyCLIP** | (85.0, 83.6) | (94.4, 84.8) | (87.7, 90.6) | (83.2, 62.9) |
| $\textbf{FAPrompt}_{0.1}$ | (87.9, 87.0) | (93.0, 82.2) | (90.6, 93.0) | (84.4, 65.1) |
| $\textbf{FAPrompt}_{0.2}$ | (87.7, 86.7) | (94.4, 83.2) | (90.9, 92.3) | (85.1, 65.7) |
| **FAPrompt** | (88.2, 87.2) | (95.0, 85.0) | (90.9, 93.0) | (85.4, 65.9) |

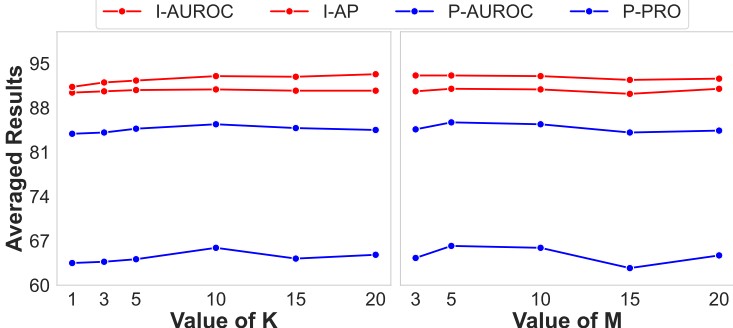

**Figure 7:** Averaged results on medical data with varying $K$ and $M$.

Given these results, we chose to average the abnormality prompts to generate the abnormality prompt prototype in `FAPrompt`, as it offers a straightforward yet effective way to integrate diverse abnormalities while preserving their complementary information.

## C.5 HYPERPARAMETER SENSITIVITY ANALYSIS

**Sensitivity Analysis for $K$ and $M$.** We present the image-level and pixel-level results for the sensitivity w.r.t. the number of abnormality prompts ($K$) in CAP and the number of selected patch tokens ($M$) across the medical datasets in DAP in Fig. 7. The trend of the results is consistent with the industrial datasets shown in Fig. 4.

**Ablation Studies on $K = 1$.** To verify the necessity of using multiple prompts, we conduct module ablation on $K = 1$ and $K = 10$. As shown by the results in Table 10, even without applying DAP, the `FAPrompt` variant using a single compound abnormality prompt '+CAP ($K = 1$)' also gains improved performance over the base model 'AnomalyCLIP'. This improvement becomes more pronounced as $k$ increases to 10, which denoted as '+CAP ($K = 10$)'. This improvement indicates that multiple prompts are effective in capturing a broader spectrum of abnormalities.

The combination of '+ CAP (k=1) + DAP' underperforms compared to using DAP alone. This is because '+ CAP (k=1) + DAP' relies on just a single abnormality prompt with a limited set of abnormal tokens, restricting its ability to capture the full diversity of abnormalities and leverage the abnormality prior provided by DAP effectively. However, when the number of abnormality prompts increases to 10, the ability of `FAPrompt` to learn diverse abnormal patterns improves substantially.

**Table 10:** Ablation study on `FAPrompt` with $K = 1$ and $K = 10$.

| Model | Industrial | | Medical | |
|---|---|---|---|---|
| | image-level | pixel-level | image-level | pixel-level |
| **AnomalyCLIP** | (85.0, 83.6) | (94.4, 84.8) | (87.7, 90.6) | (83.2, 62.9) |
| **+DAP** | (86.9, 85.2) | (94.8, 84.9) | (90.2, 92.3) | (84.6, 64.8) |
| **+ CAP** ($K = 1$) | (85.7, 85.5) | (94.5, 83.9) | (89.9, 91.3) | (83.6, 63.8) |
| **+ CAP** ($K = 1$) **+DAP** | (86.1, 86.2) | (94.3, 83.7) | (90.4, 91.3) | (83.9, 63.5) |
| **+ CAP** ($K = 10$) | (88.1, 87.0) | (94.6, 83.9) | (90.6, **93.1**) | (83.8, 63.8) |
| **+ CAP** ($K = 10$) **+DAP** | (**88.2, 87.2**) | (**95.0, 85.0**) | (**90.9**, 93.0) | (**85.4, 65.9**) |

**Table 11:** Hyperparameter analysis of the number of layers with learnable tokens and the length of the tokens.

| Model | Industrial Datasets | | Medical Datasets | |
|---|---|---|---|---|
| | Image-level | Pixel-level | Image-level | Pixel-level |
| Length of learnable token | | | | |
| 2 | (88.4, 87.4) | (95.0, 84.8) | (90.7, 91.7) | (84.9, 65.1) |
| 4 | (88.2, 87.2) | (95.0, 85.0) | (90.9, 93.0) | (85.4, 65.9) |
| 6 | (90.0, 87.7) | (94.8, 85.3) | (91.2, 93.5) | (85.0, 65.2) |
| 8 | (87.8, 86.6) | (94.9, 84.3) | (90.6, 92.3) | (85.0, 65.1) |
| Layers having learnable tokens | | | | |
| 5 | (88.0, 87.3) | (94.2, 85.5) | (91.2, 93.0) | (84.6, 65.0) |
| 7 | (88.0, 86.9) | (94.6, 84.3) | (91.0, 93.3) | (85.3, 65.2) |
| 9 | (88.2, 87.2) | (95.0, 85.0) | (90.9, 93.0) | (85.4, 65.9) |
| 11 | (88.1, 87.2) | (94.9, 84.5) | (90.5, 92.7) | (84.5, 63.5) |

As a result, '+ CAP (k=10) + DAP', which is also our full `FAPrompt`, achieves the best performance across various datasets.

These results demonstrate that using multiple prompts enables `FAPrompt` to better capture diverse, complementary abnormalities, maximizing the benefit of both CAP and DAP components for the overall superior performance.

**Sensitivity Analysis for Learnable Tokens.** To evaluate the sensitivity of the learnable tokens, we also conduct ablation studies on the number of layers with learnable tokens and the length of the tokens. As shown by the results in Table 11, the performance generally gets improved with an increasing number of layers, reaching optimal performance at 9 layers. Beyond 9 layers, it tends to over-generalization, leading to a decrease in the detection performance. A similar pattern was observed with the token length, where `FAPrompt` achieves the best overall performance with a token length of 4 and 6.

## C.6 QUALITATIVE RESULTS OF FAPROMPT

We compare the anomaly maps generated by `FAPrompt` with those produced by other ZSAD models across various datasets, as shown in Fig. 8. APRIL-GAN and AnomalyCLIP are selected as representatives of handcrafted and learnable text prompt competitors, respectively. The visualization results show that `FAPrompt` demonstrates significantly more accurate segmentation compared to the other two methods across both industrial and medical domains. In particular, despite not accessing any additional information or training from medical data, `FAPrompt` effectively localizes abnormal lesion/tumor regions, which highlight the cross-dataset generalization superiority of the fine-grained abnormality semantics learned by `FAPrompt`.

To assess the performance on samples containing multiple anomalous types within a single image, we also provide visualization of pixel-level detection results on such samples from three MVTecAD categories (zipper, pill and wood) and AITEX. The results shown in Fig. 9 demonstrate that despite using a single abnormality prompt prototype, `FAPrompt` can still effectively detect multiple anomaly types in a single image.

In addition, we also provide pixel-level anomaly score maps on diverse datasets to further showcase the strong segmentation capability of `FAPrompt` in Figs. 10 to 19. Specifically, for the industrial AD datasets, we select three object categories (capsule, pipe_fryum in VisA and metal_plate in MPDD) and three texture categories (grid, tile in MVTecAD and AITEX) for visualization. For the medical AD datasets, we visualize the pixel-level anomaly detection performance for the brain, colon, skin, and thyroid anomalies.

## C.7 FAILURE CASES AND LIMITATIONS

While the proposed `FAPrompt` demonstrates promising detection results across various categories without any dataset-specific references, it may fail in certain cases. Fig. 20 illustrates some of these failure cases. Some cases can be attributed to annotation errors. For example, images that

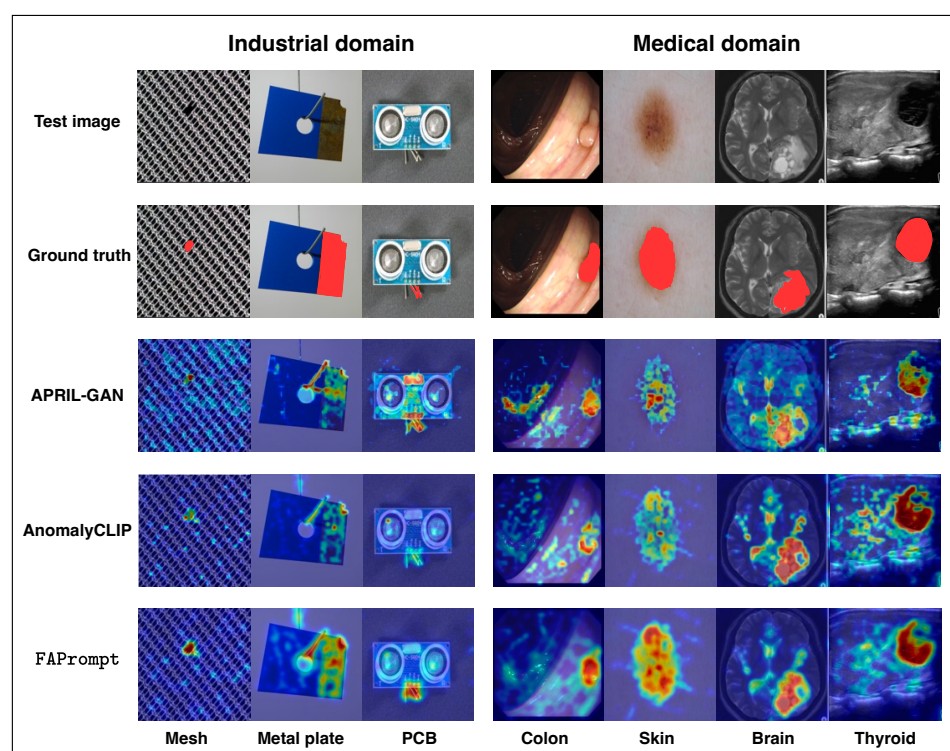

**Figure 8:** Visualization of anomaly maps generated by different ZSAD methods.

contain multiple types of anomalies but are only partially labeled may lead to segmentation errors due to labeling inconsistencies, as can be seen in the stain defect in Fig. 20 (1). Additionally, instrument artifacts in some medical datasets are often misinterpreted as anomalies, leading to incorrect detection, *e.g.*, Fig. 20 (2). In other cases, `FAPrompt` may fail in challenging cases like the ones illustrated in Fig. 20 (3)-(6), where the anomalous regions may be too small, subtle, or overshadowed by other suspicious areas (according to `FAPrompt`'s interpretation). Nevertheless, as demonstrated in this figure and Figs. 10 to 19, `FAPrompt` consistently strives to identify the most likely abnormal regions, without relying on any reference from the target datasets. Moving forward, incorporating more prior knowledge, *e.g.*, from in-context examples, knowledge graphs, or Large Language Models (LLMs), would be helpful for providing more discriminative information for achieving more accurate anomaly detection.

# D    DETAILED EMPIRICAL RESULTS

## D.1    BREAKDOWN RESULTS ON VISA AND MVTEC AD

Tables 12 to  19 present detailed downbreak ZSAD results of `FAPrompt` against eight SotA methods across each category of the MVTecAD and VisA datasets.

## D.2    DATASET-SPECIFIC RESULTS ON ABLATION STUDY

In this section, we present the dataset-specific image-level and pixel-level ZSAD results for module ablation in Table 20 and Table 21, respectively.

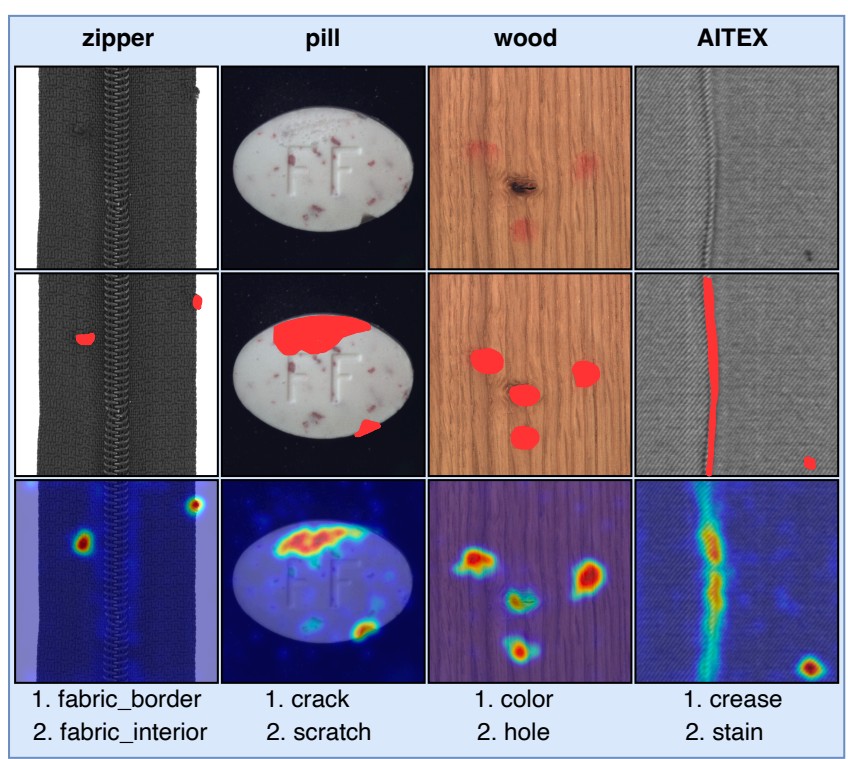

**Figure 9:** Visualization of anomaly maps of `FAPrompt` on samples containing multiple anomalous types in a single image.

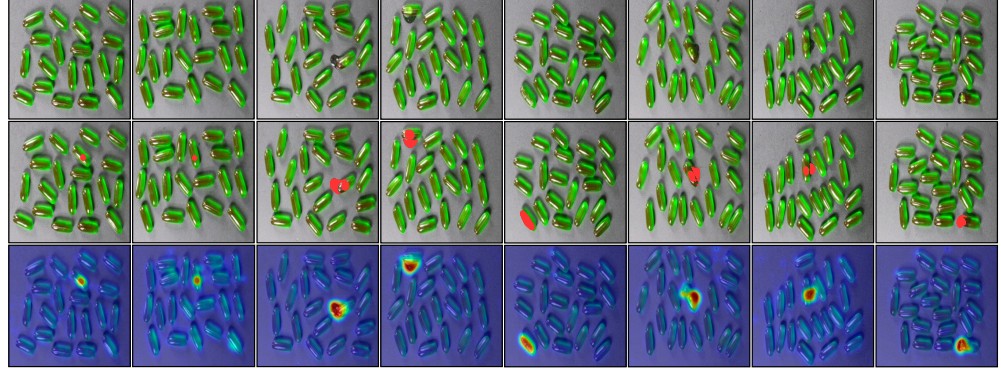

**Figure 10:** Anomaly maps generated by `FAPrompt` for the capsules category in VisA. The first row represents the input images, while the second row displays the ground truth of anomalous regions. The bottom row illustrates the segmentation results from `FAPrompt`.

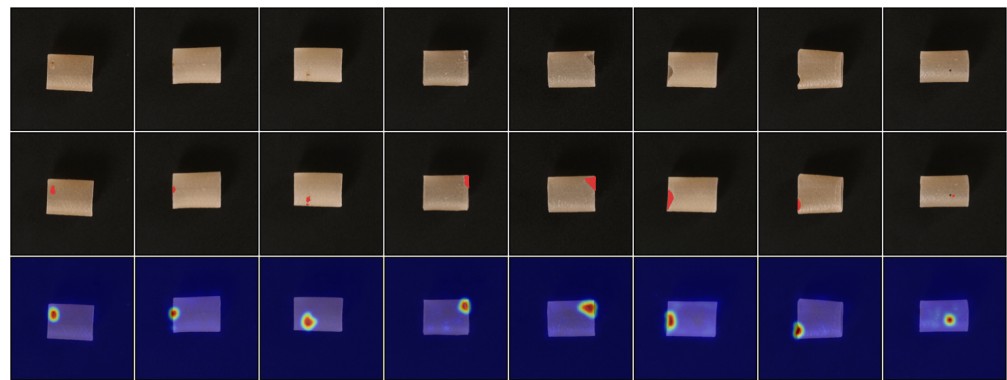

**Figure 11:** Anomaly maps generated by `FAPrompt` for the pipe_fryum category in VisA. The first row represents the input images, while the second row displays the ground truth of anomalous regions. The bottom row illustrates the segmentation results from `FAPrompt`.

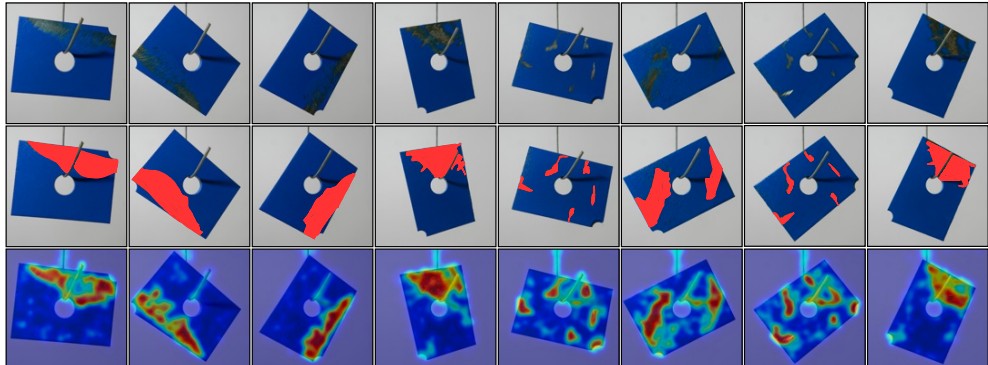

**Figure 12:** Anomaly maps generated by `FAPrompt` for the metal_plate category in MPDD. The first row represents the input images, while the second row displays the ground truth of anomalous regions. The bottom row illustrates the segmentation results from `FAPrompt`.

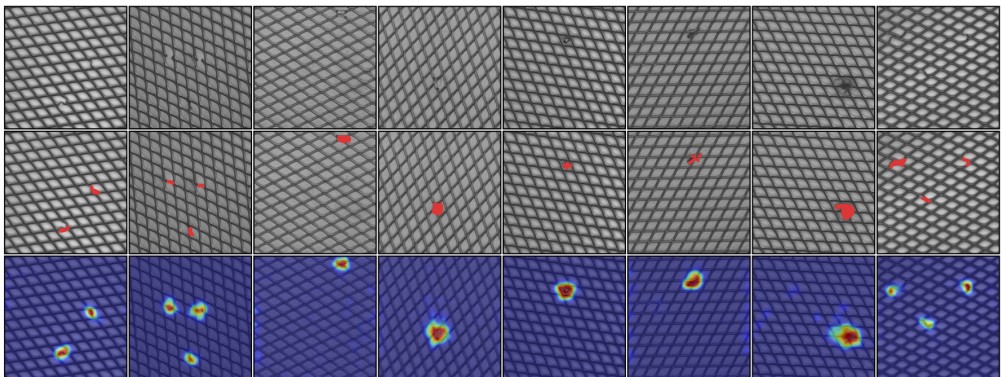

**Figure 13:** Anomaly maps generated by `FAPrompt` for grid category in MVTecAD. The first row represents the input images, while the second row displays the ground truth of anomalous regions. The bottom row illustrates the segmentation results from `FAPrompt`.

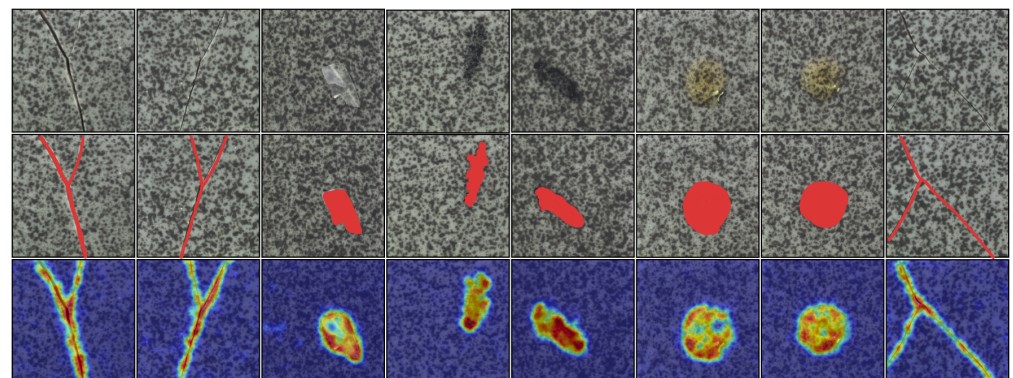

**Figure 14:** Anomaly maps generated by `FAPrompt` for tile category in MVTecAD. The first row represents the input images, while the second row displays the ground truth of anomalous regions. The bottom row illustrates the segmentation results from `FAPrompt`.

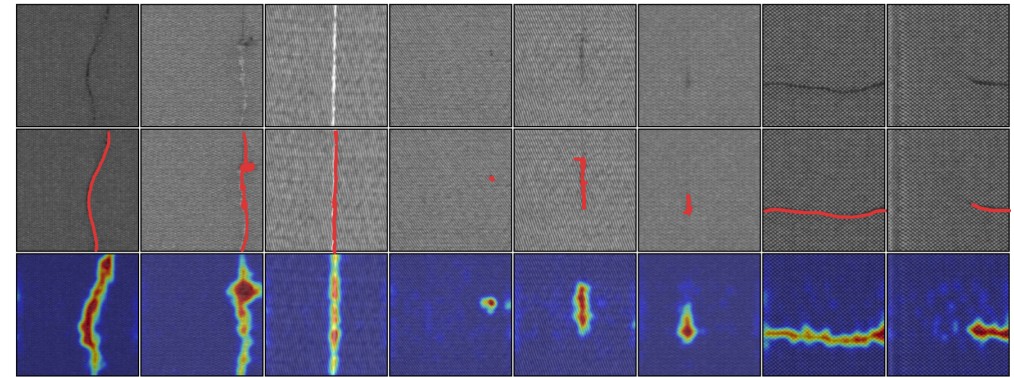

**Figure 15:** Anomaly maps generated by `FAPrompt` for AITEX. The first row represents the input images, while the second row displays the ground truth of anomalous regions. The bottom row illustrates the segmentation results from `FAPrompt`.

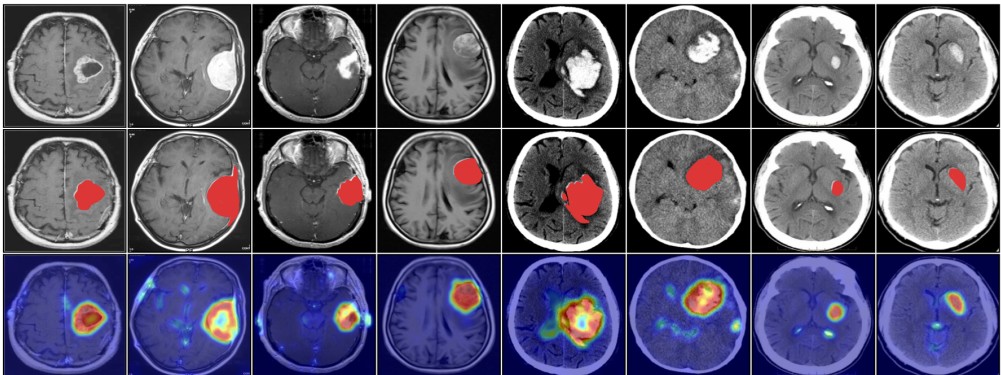

**Figure 16:** Anomaly maps generated by `FAPrompt` for brain-related anomalies. The first row represents the input images, while the second row displays the ground truth of anomalous regions. The bottom row illustrates the segmentation results from `FAPrompt`.

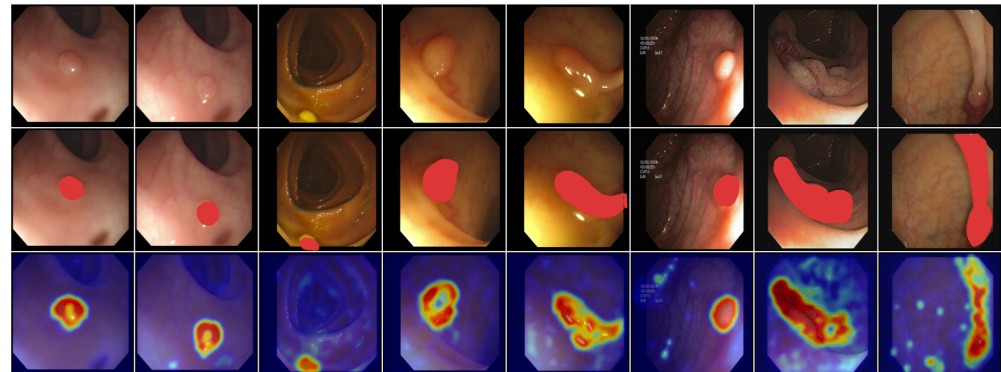

**Figure 17:** Anomaly maps generated by `FAPrompt` for colon-related anomalies. The first row represents the input images, while the second row displays the ground truth of anomalous regions. The bottom row illustrates the segmentation results from `FAPrompt`.

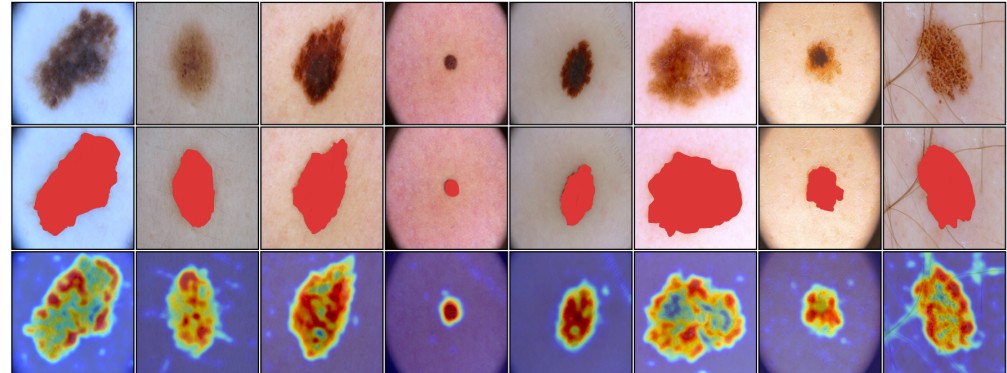

**Figure 18:** Anomaly maps generated by `FAPrompt` for skin-related anomalies. The first row represents the input images, while the second row displays the ground truth of anomalous regions. The bottom row illustrates the segmentation results from `FAPrompt`.

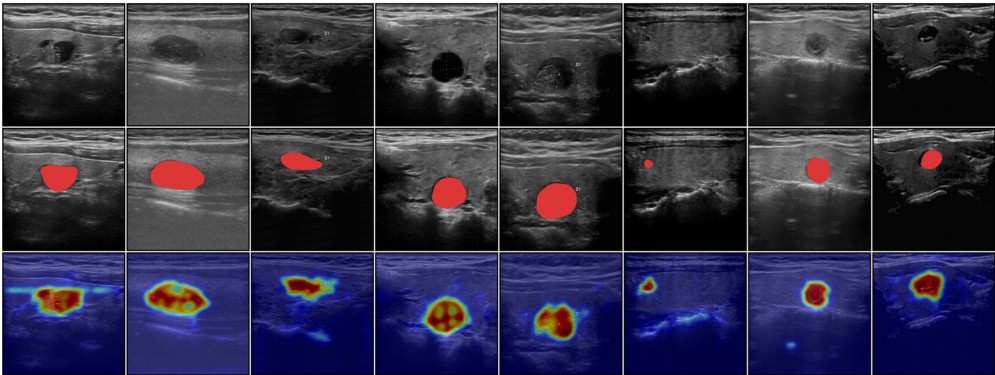

**Figure 19:** Anomaly maps generated by `FAPrompt` for thyroid-related anomalies. The first row represents the input images, while the second row displays the ground truth of anomalous regions. The bottom row illustrates the segmentation results from `FAPrompt`.

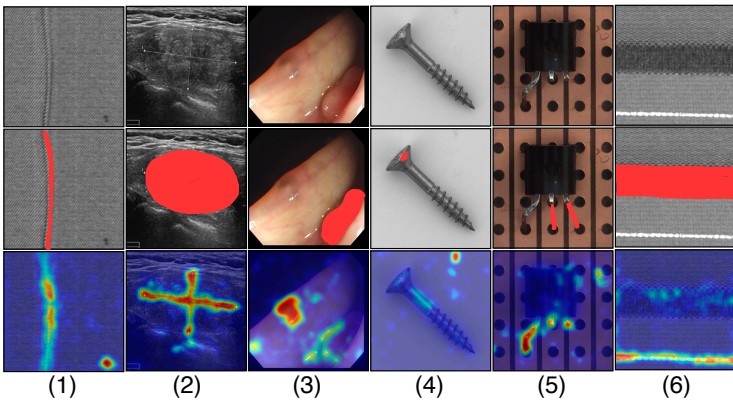

|  | (1) | (2) | (3) | (4) | (5) | (6) |

**Figure 20:** Failure cases of `FAPrompt`. The first row represents the input images, while the second row displays the ground truth of anomalous regions. The bottom row illustrates the segmentation results from `FAPrompt`.

**Table 12:** Breakdown AUROC results of image-level ZSAD performance comparison on MVTecAD.

| Data Subset | Handcrafted Text Prompting | | | | | Learnable Text Prompting | | | |
|---|---|---|---|---|---|---|---|---|---|
| | CLIP | CLIP-AC | WinCLIP | APRIL-GAN | AnoVL | CoOp | CoCoOp | AnomalyCLIP | FAPrompt |
| Carpet | 96.0 | 93.1 | 100.0 | 99.5 | - | 99.9 | 98.7 | 100.0 | 100.0 |
| Grid | 72.5 | 63.7 | 98.8 | 86.3 | - | 94.7 | 87.7 | 97.0 | 97.9 |
| Leather | 99.4 | 99.5 | 100.0 | 99.7 | - | 99.9 | 98.5 | 99.8 | 99.9 |
| Tile | 88.5 | 89.0 | 100.0 | 99.9 | - | 99.7 | 99.4 | 100.0 | 99.7 |
| Wood | 94.0 | 94.9 | 99.4 | 99.0 | - | 97.7 | 44.4 | 96.8 | 98.0 |
| Bottle | 45.9 | 46.1 | 99.2 | 92.0 | - | 87.7 | 80.2 | 89.3 | 89.8 |
| Capsule | 71.4 | 68.8 | 72.9 | 79.9 | - | 81.1 | 84.2 | 89.9 | 92.4 |
| Pill | 73.6 | 73.8 | 79.1 | 80.5 | - | 78.6 | 83.3 | 81.8 | 89.6 |
| Transistor | 48.8 | 51.2 | 88.0 | 80.8 | - | 92.2 | 77.3 | 92.8 | 81.7 |
| Zipper | 60.1 | 36.1 | 91.5 | 89.6 | - | 98.8 | 54.5 | 98.5 | 98.4 |
| Cable | 58.1 | 46.6 | 86.5 | 88.4 | - | 56.7 | 29.6 | 69.8 | 74.7 |
| Hazelnut | 88.7 | 91.1 | 93.9 | 89.6 | - | 93.5 | 11 | 97.2 | 96.5 |
| Metal_nut | 62.8 | 63.4 | 97.1 | 68.4 | - | 85.3 | 81.3 | 93.6 | 89.7 |
| Screw | 78.2 | 66.7 | 83.3 | 84.9 | - | 88.9 | 59 | 81.1 | 85.0 |
| Toothbrush | 73.3 | 89.2 | 88.0 | 53.8 | - | 77.5 | 88.6 | 84.7 | 85.6 |
| MEAN | 74.1 | 71.5 | 91.8 | 86.2 | 92.5 | 88.8 | 71.8 | 91.5 | 91.9 |

**Table 13:** Breakdown AP results of image-level ZSAD performance comparison on MVTecAD.

| Data Subset | Handcrafted Text Prompting | | | | | Learnable Text Prompting | | | |
|---|---|---|---|---|---|---|---|---|---|
| | CLIP | CLIP-AC | WinCLIP | APRIL-GAN | AnoVL | CoOp | CoCoOp | AnomalyCLIP | FAPrompt |
| Carpet | 98.8 | 97.8 | 100.0 | 99.8 | - | 100.0 | 99.6 | 100.0 | 100.0 |
| Grid | 87.1 | 83.9 | 99.6 | 94.9 | - | 98.1 | 95.8 | 99.1 | 99.3 |
| Leather | 99.8 | 99.8 | 100.0 | 99.9 | - | 100.0 | 99.3 | 99.9 | 100.0 |
| Tile | 95.9 | 96.2 | 100.0 | 100.0 | - | 99.9 | 99.8 | 100.0 | 99.9 |
| Wood | 97.9 | 98.3 | 99.8 | 99.7 | - | 99.4 | 68.2 | 99.2 | 99.4 |
| Bottle | 78.9 | 79.8 | 99.8 | 97.7 | - | 96.4 | 93.1 | 97.0 | 96.7 |
| Capsule | 92.1 | 90.9 | 91.5 | 95.5 | - | 95.7 | 96.5 | 97.9 | 98.4 |
| Pill | 93.4 | 93.6 | 95.7 | 96.0 | - | 94.2 | 96.2 | 95.4 | 97.9 |
| Transistor | 48.1 | 49.9 | 87.1 | 77.5 | - | 90.2 | 71.1 | 90.6 | 78.9 |
| Zipper | 87.4 | 73.9 | 97.5 | 97.1 | - | 99.7 | 86.7 | 99.6 | 99.5 |
| Cable | 70.8 | 64.3 | 91.2 | 93.1 | - | 69.4 | 50.8 | 81.4 | 82.9 |
| Hazelnut | 94.6 | 95.9 | 96.9 | 94.8 | - | 96.7 | 45.9 | 98.6 | 98.1 |
| Metal_nut | 87.7 | 89.2 | 99.3 | 91.9 | - | 96.3 | 93.6 | 98.5 | 97.5 |
| Screw | 91.4 | 86.6 | 93.1 | 93.6 | - | 96.2 | 81.2 | 92.5 | 93.6 |
| Toothbrush | 90.7 | 96.0 | 95.6 | 71.5 | - | 90.4 | 95.1 | 93.7 | 93.8 |
| MEAN | 87.6 | 86.4 | 96.5 | 93.5 | 96.7 | 94.8 | 84.9 | 96.2 | 95.7 |

**Table 14:** Breakdown AUROC results of pixel-level ZSAD performance comparison on MVTecAD.

| Data Subset | Handcrafted Text Prompting | | | | | Learnable Text Prompting | | | |
|---|---|---|---|---|---|---|---|---|---|
| | CLIP | CLIP-AC | WinCLIP | APRIL-GAN | AnoVL | CoOp | CoCoOp | AnomalyCLIP | FAPrompt |
| Carpet | 11.5 | 10.7 | 95.4 | 98.4 | - | 6.7 | 96.7 | 98.8 | 99.0 |
| Grid | 8.7 | 11.9 | 82.2 | 95.8 | - | 7.8 | 89.8 | 97.3 | 96.9 |
| Leather | 9.9 | 5.6 | 96.7 | 99.1 | - | 11.7 | 98.5 | 98.6 | 98.5 |
| Tile | 49.9 | 39.1 | 77.6 | 92.7 | - | 41.7 | 87.4 | 94.6 | 95.7 |
| Wood | 45.7 | 42.4 | 93.4 | 95.8 | - | 31.4 | 94.5 | 96.5 | 96.4 |
| Bottle | 17.5 | 23.3 | 89.5 | 83.4 | - | 23.1 | 89.7 | 90.4 | 90.3 |
| Capsule | 50.9 | 49.1 | 86.9 | 92.0 | - | 35.5 | 80.1 | 95.8 | 95.2 |
| Pill | 55.8 | 60.8 | 80.0 | 76.2 | - | 46.5 | 78.7 | 92.0 | 90.5 |
| Transistor | 51.1 | 48.5 | 74.7 | 62.4 | - | 50.1 | 66.2 | 71.0 | 69.8 |
| Zipper | 51.5 | 44.7 | 91.6 | 91.1 | - | 33.4 | 92.0 | 91.4 | 91.8 |
| Cable | 37.4 | 37.5 | 77.0 | 72.3 | - | 49.7 | 73.3 | 78.9 | 79.5 |
| Hazelnut | 25.2 | 34.0 | 94.3 | 96.1 | - | 30.2 | 95.9 | 97.1 | 97.5 |
| Metal_nut | 43.9 | 53.6 | 61.0 | 65.4 | - | 49.3 | 71.0 | 74.4 | 71.4 |
| Screw | 80.1 | 76.4 | 89.6 | 97.8 | - | 17.0 | 98.3 | 97.5 | 97.4 |
| Toothbrush | 36.3 | 35.0 | 86.9 | 95.8 | - | 64.9 | 89.1 | 91.9 | 89.7 |
| **MEAN** | **38.4** | **38.2** | **85.1** | **87.6** | **89.8** | **33.3** | **86.7** | **91.1** | **90.6** |

**Table 15:** Breakdown PRO results of pixel-level ZSAD performance comparison on MVTecAD.

| Data Subset | Handcrafted Text Prompting | | | | | Learnable Text Prompting | | | |
|---|---|---|---|---|---|---|---|---|---|
| | CLIP | CLIP-AC | WinCLIP | APRIL-GAN | AnoVL | CoOp | CoCoOp | AnomalyCLIP | FAPrompt |
| Carpet | 2.9 | 1.9 | 84.1 | 48.5 | - | 0.5 | 94.1 | 90.1 | 94.1 |
| Grid | 0.9 | 2.4 | 57.0 | 31.6 | - | 1.0 | 74.5 | 75.6 | 81.6 |
| Leather | 0.2 | 0.0 | 91.1 | 72.4 | - | 1.8 | 97.9 | 92.2 | 95.7 |
| Tile | 21.5 | 16.3 | 51.2 | 26.7 | - | 10.1 | 76.9 | 87.6 | 89.3 |
| Wood | 13.7 | 10.3 | 74.1 | 31.1 | - | 5.1 | 93.1 | 91.2 | 92.3 |
| Bottle | 1.4 | 4.9 | 76.4 | 45.6 | - | 4.5 | 79.4 | 80.9 | 81.0 |
| Capsule | 13.2 | 14.9 | 62.1 | 51.3 | - | 5.7 | 82.8 | 87.2 | 83.9 |
| Pill | 6.0 | 8.2 | 65.0 | 65.4 | - | 3.2 | 84.4 | 88.2 | 87.6 |
| Transistor | 15.3 | 11.2 | 43.4 | 21.3 | - | 9.3 | 51.5 | 58.1 | 59.0 |
| Zipper | 17.7 | 15.2 | 71.7 | 10.7 | - | 11.6 | 78.3 | 65.3 | 75.1 |
| Cable | 7.3 | 6.9 | 42.9 | 25.7 | - | 12.2 | 55.5 | 64.4 | 68.2 |
| Hazelnut | 2.8 | 9.4 | 81.6 | 70.3 | - | 4.7 | 89.2 | 92.4 | 93.3 |
| Metal_nut | 2.9 | 10.3 | 31.8 | 38.4 | - | 7.0 | 71.5 | 71.0 | 70.9 |
| Screw | 57.8 | 56.2 | 68.5 | 67.1 | - | 6.4 | 93.8 | 88.0 | 89.7 |
| Toothbrush | 5.8 | 5.2 | 67.7 | 54.5 | - | 16.6 | 71.6 | 88.5 | 87.3 |
| **MEAN** | **11.3** | **11.6** | **64.6** | **44.0** | **76.2** | **6.6** | **79.6** | **81.4** | **83.3** |

**Table 16:** Breakdown AUCROC results of image-level ZSAD performance comparison on VisA.

| Data Subset | Handcrafted Text Prompting | | | | | Learnable Text Prompting | | | |
|---|---|---|---|---|---|---|---|---|---|
| | CLIP | CLIP-AC | WinCLIP | APRIL-GAN | AnoVL | CoOp | CoCoOp | AnomalyCLIP | FAPrompt |
| candle | 37.9 | 33.0 | 95.7 | 83.8 | - | 46.2 | 63.7 | 79.3 | 87.2 |
| capsules | 69.7 | 75.3 | 85.0 | 61.2 | - | 77.2 | 69.8 | 81.5 | 91.6 |
| cashew | 69.1 | 72.7 | 92.2 | 87.3 | - | 75.7 | 93.3 | 76.3 | 90.5 |
| chewinggum | 77.5 | 76.9 | 95.3 | 96.4 | - | 84.9 | 96.5 | 97.4 | 97.6 |
| fryum | 67.2 | 60.9 | 75.3 | 94.3 | - | 80.0 | 76.6 | 93.0 | 96.5 |
| macaroni1 | 64.4 | 67.4 | 77.8 | 71.6 | - | 53.6 | 68.0 | 87.2 | 83.1 |
| macaroni2 | 65.0 | 65.7 | 66.7 | 64.6 | - | 66.5 | 75.4 | 73.4 | 71.4 |
| pcb1 | 54.9 | 43.9 | 79.8 | 53.4 | - | 24.7 | 81.5 | 85.4 | 68.2 |
| pcb2 | 62.6 | 59.5 | 52.6 | 71.8 | - | 44.6 | 61.6 | 62.2 | 66.4 |
| pcb3 | 52.2 | 49.0 | 70.2 | 66.8 | - | 54.4 | 66.4 | 62.7 | 68.6 |
| pcb4 | 87.7 | 89.0 | 84.5 | 95.0 | - | 66.0 | 93.8 | 93.9 | 95.4 |
| pipe_fryum | 88.8 | 86.4 | 69.4 | 89.9 | - | 80.1 | 91.0 | 92.4 | 97.4 |
| **MEAN** | **66.4** | **65.0** | **78.7** | **78.0** | **79.2** | **62.8** | **78.1** | **82.1** | **84.5** |

**Table 17:** Breakdown AP results of image-level ZSAD performance comparison on VisA.

| Data Subset | Handcrafted Text Prompting | | | | | Learnable Text Prompting | | | |
|---|---|---|---|---|---|---|---|---|---|
| | CLIP | CLIP-AC | WinCLIP | APRIL-GAN | AnoVL | CoOp | CoCoOp | AnomalyCLIP | FAPrompt |
| candle | 42.9 | 40.0 | 96.1 | 86.9 | - | 52.9 | 67.7 | 81.1 | 89.7 |
| capsules | 81.0 | 84.3 | 91.0 | 74.3 | - | 85.3 | 81.9 | 88.7 | 96.2 |
| cashew | 83.4 | 86.1 | 96.5 | 94.1 | - | 87.1 | 96.8 | 89.4 | 95.9 |
| chewinggum | 90.4 | 90.2 | 97.9 | 98.4 | - | 93.1 | 98.6 | 98.9 | 99.1 |
| fryum | 82.0 | 76.6 | 88.1 | 97.2 | - | 90.2 | 89.6 | 96.8 | 98.4 |
| macaroni1 | 56.8 | 58.7 | 77.7 | 70.9 | - | 52.3 | 73.0 | 86.0 | 82.5 |
| macaroni2 | 65.0 | 65.8 | 63.3 | 63.2 | - | 62.2 | 72.2 | 72.1 | 68.5 |
| pcb1 | 56.9 | 48.4 | 81.8 | 57.2 | - | 36.0 | 82.4 | 87.0 | 72.5 |
| pcb2 | 63.2 | 59.8 | 50.4 | 73.8 | - | 47.3 | 64.6 | 64.3 | 68.2 |
| pcb3 | 53.0 | 47.6 | 70.4 | 70.7 | - | 54.8 | 71.1 | 70.0 | 76.5 |
| pcb4 | 88.0 | 90.6 | 81.5 | 95.1 | - | 66.3 | 94.0 | 94.4 | 95.6 |
| pipe_fryum | 94.6 | 93.7 | 82.1 | 94.8 | - | 89.7 | 95.1 | 96.3 | 98.6 |
| **MEAN** | **71.4** | **70.2** | **81.4** | **81.4** | **81.7** | **68.1** | **82.3** | **85.4** | **86.8** |

**Table 18:** Breakdown AUROC results of pixel-level ZSAD performance comparison on VisA.

| Data Subset | Handcrafted Text Prompting | | | | | Learnable Text Prompting | | | |
|---|---|---|---|---|---|---|---|---|---|
| | CLIP | CLIP-AC | WinCLIP | APRIL-GAN | AnoVL | CoOp | CoCoOp | AnomalyCLIP | FAPrompt |
| candle | 33.6 | 50.0 | 88.9 | 97.8 | - | 16.3 | 97.9 | 98.8 | 98.9 |
| capsules | 56.8 | 61.5 | 81.6 | 97.5 | - | 47.5 | 89.7 | 95.0 | 96.3 |
| cashew | 64.5 | 62.5 | 84.7 | 86.0 | - | 32.5 | 85.8 | 93.8 | 95.2 |
| chewinggum | 43.0 | 56.5 | 93.3 | 99.5 | - | 3.4 | 98.5 | 99.3 | 99.3 |
| fryum | 45.6 | 62.7 | 88.5 | 92.0 | - | 21.7 | 93.3 | 94.6 | 94.4 |
| macaroni1 | 20.3 | 22.9 | 70.9 | 98.8 | - | 36.8 | 98.6 | 98.3 | 98.2 |
| macaroni2 | 37.7 | 28.8 | 59.3 | 97.8 | - | 27.5 | 99.0 | 97.6 | 96.8 |
| pcb1 | 57.8 | 51.6 | 61.2 | 92.7 | - | 19.8 | 90.4 | 94.1 | 96.0 |
| pcb2 | 34.7 | 38.4 | 71.6 | 89.7 | - | 22.9 | 89.3 | 92.4 | 92.7 |
| pcb3 | 54.6 | 44.6 | 85.3 | 88.4 | - | 18.0 | 91.3 | 88.4 | 88.2 |
| pcb4 | 52.1 | 49.9 | 94.4 | 94.6 | - | 14.0 | 93.6 | 95.7 | 97.1 |
| pipe_fryum | 58.7 | 44.7 | 75.4 | 96.0 | - | 29.2 | 96.1 | 98.2 | 98.1 |
| **MEAN** | **46.6** | **47.8** | **79.6** | **94.2** | **89.9** | **24.1** | **93.6** | **95.5** | **95.9** |

**Table 19:** Breakdown PRO results of pixel-level ZSAD performance comparison on VisA.

| Data Subset | Handcrafted Text Prompting | | | | | Learnable Text Prompting | | | |
|---|---|---|---|---|---|---|---|---|---|
| | CLIP | CLIP-AC | WinCLIP | APRIL-GAN | AnoVL | CoOp | CoCoOp | AnomalyCLIP | FAPrompt |
| candle | 3.6 | 6.0 | 83.5 | 92.5 | - | 1.1 | 92.4 | 96.2 | 95.8 |
| capsules | 15.8 | 22.4 | 35.3 | 86.7 | - | 18.4 | 72.8 | 78.5 | 84.9 |
| cashew | 9.6 | 10.9 | 76.4 | 91.7 | - | 1.7 | 93.6 | 91.6 | 90.0 |
| chewinggum | 17.8 | 30.2 | 70.4 | 87.3 | - | 0.1 | 86.1 | 91.2 | 90.1 |
| fryum | 12.1 | 29.3 | 77.4 | 89.7 | - | 2.6 | 91.3 | 86.8 | 87.1 |
| macaroni1 | 8.1 | 13.4 | 34.3 | 93.2 | - | 18.1 | 93.9 | 89.8 | 89.9 |
| macaroni2 | 20.9 | 18.4 | 21.4 | 82.3 | - | 2.7 | 89.5 | 84.2 | 80.3 |
| pcb1 | 11.7 | 12.5 | 26.3 | 87.5 | - | 0.1 | 82.1 | 81.7 | 87.3 |
| pcb2 | 12.8 | 13.9 | 37.2 | 75.6 | - | 0.7 | 72.9 | 78.9 | 77.8 |
| pcb3 | 31.7 | 23.6 | 56.1 | 77.8 | - | 0.0 | 84.6 | 77.1 | 77.8 |
| pcb4 | 17.1 | 20.3 | 80.4 | 86.8 | - | 0.0 | 84.8 | 91.3 | 91.7 |
| pipe_fryum | 16.7 | 6.0 | 82.3 | 90.9 | - | 0.6 | 96.2 | 96.8 | 97.2 |
| **MEAN** | **14.8** | **17.2** | **56.8** | **86.8** | **71.2** | **3.8** | **86.7** | **87.0** | **87.5** |

**Table 20:** Dataset-specific image-level ZSAD results (AUROC, AP) of our ablation study.

| Data type | Dataset | Base | CAP | CAP w\o $\mathcal{L}_{oc}$ | DAP | DAP w\o $\mathcal{L}_{prior}$ | FAPrompt |
|---|---|---|---|---|---|---|---|
| Object | VisA | (82.1, 85.4) | (83.8, 86.7) | (83.8, 86.7) | (82.7, 85.0) | (81.0, 83.3) | (84.5, 86.8) |
| | SDD | (98.1, 93.4) | (98.6, 96.1) | (98.0, 95.8) | (98.1, 95.5) | (98.3, 95.3) | (98.6, 95.9) |
| | BTAD | (88.3, 87.3) | (91.5, 92.4) | (90.8, 91.1) | (90.7, 90.7) | (91.0, 89.3) | (92.0, 92.2) |
| | MPDD | (77.0, 82.0) | (78.7, 81.3) | (77.9, 81.3) | (74.6, 78.3) | (73.4, 77.8) | (80.6, 83.3) |
| Textual | AITEX | (62.2, 40.4) | (72.8, 55.8) | (72.7, 75.4) | (73.6, 54.1) | (75.9, 57.8) | (71.9, 53.2) |
| | DAGM | (97.5, 92.3) | (97.9, 93.0) | (97.9, 93.0) | (96.5, 88.2) | (95.7, 89.6) | (98.9, 95.7) |
| | DTD-Synthetic | (93.5, 97.0) | (96.3, 98.5) | (95.7, 97.0) | (96.0, 98.0) | (96.3, 98.1) | (95.9, 98.3) |
| | ELPV | (81.5, 91.3) | (84.8, 92.6) | (80.8, 90.7) | (83.0, 91.6) | (80.6, 89.9) | (83.5, 92.0) |
| Medical | BrainMRI | (90.3, 92.2) | (95.2, 95.2) | (95.0, 94.6) | (95.9, 96.0) | (95.9, 96.5) | (95.5, 95.6) |
| | HeadCT | (93.4, 91.6) | (94.7, 94.6) | (93.7, 90.4) | (92.3, 90.4) | (92.0, 91.0) | (94.8, 93.5) |
| | LAG | (74.3, 84.9) | (75.2, 85.4) | (75.2, 85.4) | (75.2, 85.5) | (74.5, 84.6) | (75.6, 85.4) |
| | Br35H | (94.6, 94.7) | (97.4, 97.1) | (97.1, 96.8) | (97.3, 97.1) | (97.0, 96.9) | (97.8, 97.5) |

**Table 21:** Dataset-specific pixel-level ZSAD results (AUROC, PRO) of our ablation study.

| Data type | Dataset | Base | CAP | CAP w\o $\mathcal{L}_{oc}$ | DAP | DAP w\o $\mathcal{L}_{prior}$ | FAPrompt |
|---|---|---|---|---|---|---|---|
| Object | VisA | (95.5, 87.0) | (95.1, 85.1) | (95.1, 85.0) | (95.8, 86.1) | (95.6, 85.1) | (95.9, 87.5) |
| | SDD | (98.1, 95.2) | (98.3, 93.8) | (98.3, 93.2) | (97.9, 95.6) | (97.7, 92.5) | (98.3, 93.6) |
| | BTAD | (94.2, 74.8) | (94.4, 70.5) | (94.4, 70.5) | (95.4, 73.7) | (95.5, 75.2) | (95.6, 75.2) |
| | MPDD | (96.5, 87.0) | (95.9, 86.2) | (95.9, 86.2) | (95.8, 86.4) | (95.5, 85.4) | (96.5, 87.9) |
| Textual | AITEX | (83.0, 66.5) | (82.3, 64.5) | (81.3, 61.9) | (82.4, 65.2) | (82.0, 62.1) | (82.0, 62.6) |
| | DAGM | (95.6, 91.0) | (98.1, 95.2) | (97.5, 95.2) | (98.5, 96.0) | (98.2, 94.4) | (98.3, 95.4) |
| | DTD-Synthetic | (97.9, 92.3) | (97.9, 92.3) | (97.9, 92.3) | (98.1, 91.4) | (98.1, 91.3) | (98.3, 93.1) |
| Medical | CVC-ColonDB | (81.9, 71.3) | (83.7, 72.8) | (82.9, 68.1) | (83.8, 73.9) | (84.0, 73.0) | (84.6, 74.7) |
| | CVC-ClinicDB | (82.9, 67.8) | (83.2, 67.8) | (83.4, 72.9) | (83.6, 68.4) | (83.3, 68.3) | (84.7, 70.1) |
| | Kvasir | (78.9, 45.6) | (78.8, 48.1) | (78.5, 48.0) | (79.3, 45.5) | (79.0, 45.3) | (81.2, 47.8) |
| | Endo | (84.1, 63.6) | (84.3, 63.4) | (84.1, 63.4) | (84.7, 63.8) | (84.8, 64.2) | (86.4, 67.2) |
| | ISIC | (89.7, 78.4) | (88.7, 78.0) | (88.1, 76.8) | (91.0, 80.9) | (91.4, 81.3) | (90.9, 81.2) |
| | TN3K | (81.5, 50.4) | (84.2, 52.7) | (84.5, 53.4) | (84.9, 56.0) | (84.2, 53.5) | (84.5, 54.1) |

