# OpenReview forum: "Fine-grained Abnormality Prompt Learning for Zero-shot Anomaly Detection"
_ICLR.cc/2025/Conference — Submitted to ICLR 2025_

### Official Review · Reviewer_RtD8 · 2024-10-27

**Soundness:** 3
**Presentation:** 4
**Contribution:** 2
**Rating:** 5
**Confidence:** 4

**Summary:**

The paper focuses on zero-shot anomaly detection by learning a set of fine-grained abnormality prompts based on CLIP. These abnormality prompts consist of shared normal tokens and specific abnormal tokens, providing fine-grained semantics of abnormality. To account for dataset variance, the authors further propose a data-dependent abnormality prior (DAP) to generate instance-aware prompts from the image patch features. Extensive experiments are conducted on 19 datasets, covering both industrial and medical scenarios, comparing the proposed method with a sufficient number of zero-shot anomaly detectors.

**Strengths:**

1. The design of compounding normal-abnormal tokens is interesting and novel.
2. The experiments are conducted extensively and demonstrate favorable performance.

**Weaknesses:**

1. The design of abnormality prompts is not entirely consistent with the concept of "fine-grained." Although CAP leverages a set of abnormality prompts with an orthogonal constraint loss, the final abnormality embedding is derived as the average of all abnormality prompt embeddings, resulting in a vector-based abnormality prompt prototype that essentially reduces the diversity of abnormalities. If there are multiple types of anomalies within an image, is a single abnormality prompt prototype sufficient?
2. The number of abnormality prompts, K, shows very limited difference to pixel-level detection results (see Figure 7), making it difficult to evaluate the necessity of multiple abnormality prompts. Additionally, Figures 4 and 7 are missing the results when K=1.
3. The proposed DAP is technically similar to CoCoOp, and it uses the features of selected image patches instead of the global image feature. Moreover, the parameter M appears to be quite sensitive for pixel-level ZSAD.

**Questions:**

1. I have some doubts about L087-L089. Since abnormality prompts have a different number of tokens compared to normal prompts, does this not make them easily distinguishable from normal prompts?
2. The first contribution is not accurate. The update of prompts indeed depends on human annotations in the cross-dataset scenario.
3. I suggest moving Figure 7 to the main manuscript, as both image-level and pixel-level zero-shot anomaly detection (ZSAD) are equally important for a comprehensive evaluation.

---

> ### Author Response · Authors · 2024-11-22
> **Responses to Reviewer RtD8**
>
> We sincerely thank your valuable and constructive feedback and suggestions. Below, we provide a detailed, point-by-point response to your questions, and hope these could help address your concerns.
>
> > #### **Weakness \#1. "The design of abnormality prompts is not entirely consistent with the concept of "fine-grained." Although CAP leverages a set of abnormality prompts with an orthogonal constraint loss, the final abnormality embedding is derived as the average of all abnormality prompt embeddings, resulting in a vector-based abnormality prompt prototype that essentially reduces the diversity of abnormalities. If there are multiple types of anomalies within an image, is a single abnormality prompt prototype sufficient?"**
>
> Please refer to our response to Shared Concern \#5 in the **General Response** section above for detailed clarification of why the averaging operation is used in CAP.
>
> To assess the performance on samples containing multiple anomalous types within a single image, we also provide visualization of pixel-level detection results on such samples from three MVTecAD categories (zipper, pill and wood) and AITEX. The results shown in Figure 9, page 24 (See $\texttt{Appendix}$ C.6) in the revised manuscript demonstrate that despite using a single abnormality prompt prototype, $\texttt{FAPrompt}$ can still effectively detect multiple anomaly types in a single image.
>
> > #### **Weakness \#2. "The number of abnormality prompts, K, shows very limited difference to pixel-level detection results (see Figure 7), making it difficult to evaluate the necessity of multiple abnormality prompts. Additionally, Figures 4 and 7 are missing the results when K=1."**
>
> Please refer to our response to Shared Concern \#4 in the **General Response** section above for results of $\texttt{FAPrompt}$ with $K=1$. We also add the results with $K=1$ into Figures 4 and 7 in our revised manuscript.
>
> As illustrated by the updated Figures 4 and 7, we observe noticeable differences in pixel-level metrics, particularly in AUPRO, with varying $K$ values. This supports the importance of using multiple prompts for fine-grained abnormality detection, since increasing $K$ improves the model's ability for more diverse abnormality learning and better detection performance at the pixel level.
>
> > #### **Weakness \#3. "The proposed DAP is technically similar to CoCoOp, and it uses the features of selected image patches instead of the global image feature. Moreover, the parameter M appears to be quite sensitive for pixel-level ZSAD."**
>
> Please refer to our response to Shared Concern \#1 in the **General Response** section above for clarification of difference between DAP and CoCoOp, and the sensitivity analysis of $M$.

---

> > ### Author Response · Authors · 2024-11-22
> > **Responses to Reviewer RtD8**
> >
> > > #### **Question \#1. "I have some doubts about L087-L089. Since abnormality prompts have a different number of tokens compared to normal prompts, does this not make them easily distinguishable from normal prompts?"**
> >
> > The compound design of abnormal tokens on top of the normal prompt in the abnormality prompts is grounded in the observation that abnormal samples often exhibit varying magnitudes of deviation from their normal counterparts while still belonging to the same class (See Line 221-224). By maintaining class-level consistency on the normal tokens and encouraging orthogonality among the abnormality prompts, this approach learns abnormality prompts that capture diverse fine-grained information about the abnormalities. This ensures that the prompts are sensitive to subtle deviations without losing the context of the class to which they belong.
> >
> > Furthermore, if abnormality prompts are too distant from the normal prompt, it would be difficult for them to provide complementary discriminative power. The compound prompt design helps enforce close yet separable proximity from the abnormal prompts to the normal prompt, enabling more effective learning of the complementary, discriminative abnormalities.
> >
> > > #### **Question \#2. "The first contribution is not accurate. The update of prompts indeed depends on human annotations in the cross-dataset scenario."**
> >
> > We apologize for any confusion regarding the human annotations. The detailed human annotations' mentioned in Lines 103-104 of original manuscript refer to the detailed text descriptions of normal and abnormal samples or fine-grained anomaly categories, instead of the binary ground-truth labels or masks. We have clarifies this part in the revised manuscript (Lines 105-106) to prevent further misunderstanding.
> >
> > > #### **Question \#3. "I suggest moving Figure 7 to the main manuscript, as both image-level and pixel-level zero-shot anomaly detection (ZSAD) are equally important for a comprehensive evaluation."**
> >
> > Thank you for the suggestion. We have included both image-level and pixel-level results on industrial datasets to the main text in Figure 4 in the revised manuscript. The results on medical image datasets are presented in Figure 7 in Appendix C.5.

---

> ### Author Response · Authors · 2024-11-25
>
> Dear Reviewer RtD8,
>
> We would like to gently remind you of our recent response. If we have sufficiently addressed your questions, we would be grateful if you could consider updating your rating in your review.
>
> If you have any additional questions or need further clarification, please feel free to let us know. We are more than happy to provide any additional information you might need. Thank you again for your time and valuable feedback.
>
> Authors

---

### Official Review · Reviewer_agpf · 2024-10-28

**Soundness:** 3
**Presentation:** 3
**Contribution:** 2
**Rating:** 5
**Confidence:** 4

**Summary:**

The paper introduces FAPrompt, a novel framework designed to improve ZSAD by enabling fine-grained detection of abnormalities without requiring dataset-specific training or detailed human annotations. Unlike existing ZSAD methods that primarily capture coarse abnormality features, FAPrompt focuses on recognizing diverse, specific abnormality details across a range of datasets, such as industrial defects and medical anomalies.

**Strengths:**

1. FAPrompt addresses a key limitation in current ZSAD methods, which often struggle to identify fine-grained, specific abnormalities. By introducing learnable fine-grained abnormality prompts, FAPrompt improves both the accuracy and applicability of ZSAD in diverse contexts.
2. Comprehensive experiments show that FAPrompt outperforms state-of-the-art methods by notable margins (3%-5% improvements in AUC/AP) across both image- and pixel-level ZSAD tasks.

**Weaknesses:**

1. The novelty of FAPrompt is limited. The design of CAP seems a simple combination of prompt tuning and prototype learning.
2. Besides CoOp and CoCoOp, other state-of-the-art prompt tuning approaches are expected for comparisons (e.g. PromptSRC[1] and TCP[2]).

*Reference*

[1] Khattak, Muhammad Uzair, et al. "Self-regulating prompts: Foundational model adaptation without forgetting." Proceedings of the IEEE/CVF International Conference on Computer Vision. 2023.

[2] Yao, Hantao, Rui Zhang, and Changsheng Xu. "TCP: Textual-based Class-aware Prompt tuning for Visual-Language Model." Proceedings of the IEEE/CVF Conference on Computer Vision and Pattern Recognition. 2024.

**Questions:**

1. In table 4, image-level results on medical datasets show that CAP achieves the best result compared with FAPrompt, is there any further explanation?
2. How to demonstrate that CAP's ability of learning fine-grained abnormality semantics? In fig 3, fine-grained prompts (with different colors) seems not to be seperable.

---

> ### Author Response · Authors · 2024-11-22
> **Responses to Reviewer agpf**
>
> We sincerely thank your valuable and constructive feedback and suggestions. Below, we provide a detailed, point-by-point response to your questions, and hope these could help address your concerns.
>
> > #### **Weaknesses \#1. "The novelty of $\texttt{FAPrompt}$ is limited. The design of CAP seems a simple combination of prompt tuning and prototype learning."**
>
> Our work introduces a new ZSAD framework that devises two novel modules -- the CAP and DAP modules -- to learn fine-grained abnormalities. As far as we know, there is no similar work done on this problem. The CAP module differs significantly from existing prompt tuning methods in terms of both intuition and design:
>
> - **Intuition.** The compound abnormality design in CAP is based on the insight that abnormal samples exhibit varying magnitudes of deviation from their normal counterparts (see Lines 221-224). Thus, it is essential for having prompts that are able to capture the varying abnormalities for effective ZSAD. CAP is designed specifically for this purpose. In contrast, previous prompt tuning-based AD methods are focused on single abnormality prompt learning, and thus, do not account for this characteristic of abnormalities.
> - **Design.** To learn diverse, complementary abnormalities, we introduce a set of compound abnormality prompts in CAP, guided by an orthogonal constraint on the prompts. This helps effectively encourage the learning complementary abnormality prompts, enforcing that they capture distinct aspects of abnormalities. This effect is illustrated by visualization results in Figure 3 and Figure 6 in the revised manuscript. Please also refer to our response to Shared Concern \#2 in the **General Response** section for a clarification on the significance of the compound prompt design). As far as we know, this design is unique and novel in the AD and other topics.
>
> In addition to CAP, the DAP module also has major novelty. Please refer to our response to Shared Concern \#1 in the **General Response** section above for the innovation of DAP.
>
> > #### **Weakness \#2. "Besides CoOp and CoCoOp, other state-of-the-art prompt tuning approaches are expected for comparisons (e.g. PromptSRC[1] and TCP[2])."**
>
> Both PromptSRC and TCP are not originally designed for Anomaly Detection, and their contextual information relies heavily on handcrafted text prompts. Due to time constraints during the discussion period, we focused on conducting additional experiments with the more recent TCP approach.
>
> For a fair comparison, we maintained the original design and settings of TCP throughout the experiments.
> Specifically, we adapted TCP for the ZSAD by testing two types of AD-oriented text prompts, resulting in two variants of TCP for ZSAD: **TCP\_V1** and **TCP\_V2**:
>
> - **TCP\_V1**, where we used a straightforward prompt design:
>
>   Normal prompt: "This is a photo of [cls]."; Abnormal prompt: "This is a photo of damaged [cls].
>
> - **TCP\_V2**, where we adopted the complete set of the prompt templates from WinCLIP.
>
> The results are shown in the table below:
>
> |        Model        | Industrial Datasets  |                      |   Medical Datasets   |                      |
> | :-----------------: | :------------------: | :------------------: | :------------------: | :------------------: |
> |                     |   **Image-level**    |   **Pixel-level**    |   **Image-level**    |   **Pixel-level**    |
> |   **AnomalyCLIP**   |     (85.0, 83.6)     |     (94.4, 84.8)     |     (87.7, 90.6)     |     (83.2, 62.9)     |
> |     **TCP\_V1**     |     (61.3, 55.9)     |     (87.2, 66.6)     |     (56.4, 61.7)     |     (80.2, 60.9)     |
> |     **TCP\_V2**     |     (64.9, 59.1)     |     (88.5, 71.5)     |     (53.3, 60.3)     |     (76.8, 52.9)     |
> | $\texttt{FAPrompt}$ | (**88.2**, **87.2**) | (**95.0**, **85.0**) | (**90.9**, **93.0**) | (**85.4**, **65.9**) |
>
> As shown, both TCP variants largely underperform AnomalyCLIP and $\texttt{FAPrompt}$ in the ZSAD task. This is primarily due to the fact that TCP is not designed for ZSAD and also has strong reliance on handcrafted text prompts.
>
> In contrast, $\texttt{FAPrompt}$ is specifically designed for the ZSAD task, leveraging data-dependent abnormality prior of the query images to learn complementary abnormality prompts. This adaptive approach enables $\texttt{FAPrompt}$ to more effectively capture a wide variety of anomalies, resulting in promising performance in both image-level and pixel-level ZSAD tasks. We have included these results in $\texttt{Appendix}$ C.2 to address this concern.
>
> **References**
>
> [1] Khattak, Muhammad Uzair, et al. "Self-regulating prompts: Foundational model adaptation without forgetting." Proceedings of the IEEE/CVF International Conference on Computer Vision. 2023.
>
> [2] Yao, Hantao, Rui Zhang, and Changsheng Xu. "TCP: Textual-based Class-aware Prompt tuning for Visual-Language Model." Proceedings of the IEEE/CVF Conference on Computer Vision and Pattern Recognition. 2024.

---

> ### Author Response · Authors · 2024-11-22
> **Responses to Reviewer agpf**
>
> > #### **Question \#1. "In table 4, image-level results on medical datasets show that CAP achieves the best result compared with $\texttt{FAPrompt}$, is there any further explanation?"**
>
> The AP metric is sensitive to detection precision and recall. In smaller datasets, a single incorrect or correct detection can significantly influence these two metrics, causing more variability in AP. Our breakdown ablation study results (Table 17) in $\texttt{Appendix}$ D.2 provides a comprehensive analysis of image-level results on four different medical datasets. Notably, the difference in AP between CAP and $\texttt{FAPrompt}$ is observed primarily in the HeadCT dataset, which is small with only 200 samples (further details can be found in the $\texttt{Appendix}$ A.1). In this dataset, '+ CAP' achieves an AP of 94.6, while $\texttt{FAPrompt}$ achieves an AP of 93.5. However, in the other three datasets, including BrainMRI (253 samples), LAG (2,497 samples) and Br35H (3,000 samples), $\texttt{FAPrompt}$ consistently outperforms '+ CAP'.
>
> > #### **Question \#2. "How to demonstrate that CAP's ability of learning fine-grained abnormality semantics? In fig 3, fine-grained prompts (with different colors) seems not to be seperable."**
>
> Please refer to our response to Shared Concern \#3 in the **General Response** section above for clarification and empirical evidence on the ability of learning fine-grained abnormalities.

---

> ### Author Response · Authors · 2024-12-02
>
> Dear Reviewer **agpf**,
>
> The author-reviewer discussion will end soon. Please kindly check whether our response helps address your concerns or not. We're more than happy to provide more details or empirical justification if you have any follow-up questions. Thank you!
>
> Best regards,
>
> Authors of Paper 9942

---

### Official Review · Reviewer_nnbo · 2024-11-03

**Soundness:** 2
**Presentation:** 2
**Contribution:** 3
**Rating:** 5
**Confidence:** 4

**Summary:**

This study proposes FAPrompt to learn fine-grained abnormalities in Zero-shot Anomaly Detection (ZSAD) and classify anomalies based on data-dependent characteristics. To address the limitations of focusing on coarse-grained abnormality semantics, the authors introduce compound abnormality prompting (CAP) and a data-dependent abnormality prior (DAP) module, along with distinct loss terms to effectively train the CAP and DAP modules.

**Strengths:**

The motivation of this study is reasonable, and the proposed method is novel. Additionally, the method has been validated across diverse datasets.

**Weaknesses:**

1. The description of the learnable layers in the DAP module is missing.
2. The visualization in Figure 1 of the proposed methodology conflicts with the actual experimental results shown.

**Questions:**

1. Please provide specific examples or criteria that distinguish 'coarse-grained' prompts from the proposed fine-grained prompts. Clarifying this would help determine if the label is used due to a single prompt approach or if there is a deeper rationale.
2. Consider alternatives to averaging the abnormality prompts in the CAP module, such as selecting the most similar prompt for each detected abnormality. If other approaches were tested, sharing the rationale or results behind choosing the current method would add clarity.
3. Could the authors provide additional evidence or clarification on how the fine-grained distributions in Figures 1 and 3 align with each other? Additional context or visual consistency would strengthen the claim that each abnormality prompt represents a specific abnormal state.
4. The explanation regarding the effectiveness of the compound prompt seems insufficient. Could more information be provided on the outcomes when combining the normal learnable token with the abnormality token? Specifically, what results were achieved if CAP and DAP were applied independently, without this combination?
5. Could the authors clarify how this approach differs from existing prompt ensemble methods? Specifically, what distinguishes the process of averaging the abnormality prompts from a traditional ensemble approach, and in what ways does this differ from selecting the highest anomaly score across individual abnormality prompts?

---

> ### Author Response · Authors · 2024-11-22
> **Responses to Reviewer nnbo**
>
> We sincerely thank your valuable and constructive feedback and suggestions. Below, we provide a detailed, point-by-point response to your questions, and hope these could help address your concerns.
>
> > #### **Weaknesses \#1. "The description of the learnable layers in the DAP module is missing."**
>
> The original manuscript has included the description of the learnable layers in DAP, which is also the abnormality prior network $\psi(\cdot)$ (See Line 274-276). We also provide the network details of $\psi(\cdot)$ in Appendix B.1.
>
> > #### **Weaknesses \#2. "The visualization in Figure 1 of the proposed methodology conflicts with the actual experimental results shown."**
>
> Thank you for your insightful feedback. We apologize for any confusion caused by Figure 1. The diversity of fine-grained abnormality prompts in our method is learned through the compound abnormality prompting and an orthogonal constraint loss, rather than by relying on additional human annotations for specific types of anomalies. Our intention is not to claim that each individual prompt represents exactly a particular anomaly type. Instead, our goal is to encourage diversity and complementarity among the learned fine-grained abnormality prompts. To address this issue, we have revised the data visualization in Figure 1 and clarify the explanation in the revised manuscript to prevent the potential confusions. This visualization is now consistent with our observations from other qualitative results, such as those in Figure 3.

---

> ### Author Response · Authors · 2024-11-22
> **Responses to Reviewer nnbo**
>
> > #### **Questions \#1. "Please provide specific examples or criteria that distinguish 'coarse-grained' prompts from the proposed fine-grained prompts. Clarifying this would help determine if the label is used due to a single prompt approach or if there is a deeper rationale."**
>
> Thank you very much for the suggestion. Please refer to our response to Shared Concern \#3 in the **General Response** section above for clarification and empirical evidence on the ability of $\texttt{FAPrompt}$ in learning fine-grained abnormalities.
>
> > #### **Questions \#2. "Consider alternatives to averaging the abnormality prompts in the CAP module, such as selecting the most similar prompt for each detected abnormality. If other approaches were tested, sharing the rationale or results behind choosing the current method would add clarity."**
> >
> > #### **and Questions \#5. "Could the authors clarify how this approach differs from existing prompt ensemble methods? Specifically, what distinguishes the process of averaging the abnormality prompts from a traditional ensemble approach, and in what ways does this differ from selecting the highest anomaly score across individual abnormality prompts."**
>
> Thank you very much for the suggestion. Please refer to our response to Shared Concern \#5 in the **General Response** section above for detailed clarification and empirical evidence on alternative methods comparing to the averaging method in CAP.
>
> > #### **Questions \#3. "Could the authors provide additional evidence or clarification on how the fine-grained distributions in Figures 1 and 3 align with each other?Additional context or visual consistency would strengthen the claim that each abnormality prompt represents a specific abnormal state."**
>
> We apologize for the misleading that `each abnormality prompt represents a specific abnormal state' implied in Figure 1. We aim to highlight throughout the paper except Figure 1 that $\texttt{FAPrompt}$ learns a set of complementary, fine-grained abnormality prompts. Indeed, each abnormality prompt learned in $\texttt{FAPrompt}$ does not correspond to a specific abnormal state/type. This is very challenging to achieve without detailed human annotations of abnormality on specific abnormal states/types (recall that $\texttt{FAPrompt}$ and its competing methods are trained with coarse binary anomaly labels only). Please refer to our response to Weakness \#2 for the consistency between Figures 1 and 3.
>
> > #### **Questions \#4. "The explanation regarding the effectiveness of the compound prompt seems insufficient. Could more information be provided on the outcomes when combining the normal learnable token with the abnormality token? Specifically, what results were achieved if CAP and DAP were applied independently, without this combination?"**
>
> The original manuscript has included ablation studies where CAP and DAP were applied independently (see Table 3). The results demonstrate that both CAP and DAP independently contribute to improved performance over the base model, and the full model $\texttt{FAPrompt}$ achieves its best results.
>
> This conclusion also holds when the number of abnormality prompts changes. Please refer to our response to Shared Concern \#4 in the **General Response** section above for detailed clarification.
>
> The effectiveness of our compound abnormality prompts is also verified by comparing $\texttt{FAPrompt}$ to the prompt ensemble methods based on AnomalyCLIP. Please our response to Shared Concern \#2 in the **General Response** section above for detailed clarification on this.

---

> > ### Author Response · Authors · 2024-12-02
> >
> > Dear Reviewer **nnbo**,
> >
> > The author-reviewer discussion will end soon. Please kindly check whether our response helps address your concerns or not. We're more than happy to provide more details or empirical justification if you have any follow-up questions. Thank you!
> >
> > Best regards,
> >
> > Authors of Paper 9942

---

### Official Review · Reviewer_TTqm · 2024-11-03

**Soundness:** 3
**Presentation:** 4
**Contribution:** 2
**Rating:** 5
**Confidence:** 3

**Summary:**

The paper introduces a novel framework for zero-shot anomaly detection (ZSAD) that focuses on learning fine-grained abnormality prompts without requiring detailed human annotations or text descriptions. Key contributions include the development of a Compound Abnormality Prompting (CAP) module for generating complementary abnormality prompts and a Data-dependent Abnormality Prior (DAP) module aimed at enhancing cross-dataset generalization. The authors assert that their method, FAPrompt, significantly outperforms existing state-of-the-art solutions in both image- and pixel-level ZSAD tasks across 19 real-world datasets.

**Strengths:**

1. The paper is clearly written and well-organized, making complex ideas accessible. Diagrams and figures effectively illustrate key ideas, enhancing reader comprehension.
2. The experiments are thorough, encompassing 19 diverse datasets from both industrial and medical domains. The results demonstrate substantial improvements over current state-of-the-art methods, reflecting the high quality of the proposed approach.

**Weaknesses:**

1. There is marginal improvement in pixel-level ZSAD in Table 2. In contrast, the simple AnomalyCLIP achieves comparable, and even superior, results on industrial datasets.
2. The design of DAP is very similar to CoCoOp, and using a fixed M across images with varying scales of anomalous regions is unreasonable.
3. Missing necessary baseline: AnomalyCLIP with an ensemble of multiple abnormality prompts with orthogonal constraint loss; otherwise, it is difficult to justify the fine-grained prompts.

**Questions:**

1. Could you provide additional evidence to substantiate the claim that the prompt-wise anomaly scores visualized in Figure 3 demonstrate the discriminability of FAPrompt? A comparison with visualizations from baseline models would effectively highlight the advantages.
2. To better verify the necessity of using multiple prompts, it is suggested to include more detailed ablation experiments, such as k=1 with and without CAP/DAP, similar to those in Table 4.
3. Ablation studies on the number of layers with learnable tokens and the length of the tokens should be included.
4. Minor: I recommend merging Table 3 and Table 4 for improved method comparison.

---

> ### Author Response · Authors · 2024-11-22
> **Responses to Reviewer TTqm**
>
> We sincerely thank your valuable and constructive feedback and suggestions. Below, we provide a detailed, point-by-point response to your questions, and hope these could help address your concerns.
>
> > #### **Weakness \#1. "There is marginal improvement in pixel-level ZSAD in Table 2. In contrast, the simple AnomalyCLIP achieves comparable, and even superior, results on industrial datasets."**
>
> Thank you very much for the comment, but it misses various important empirical results.
>
> It is true that the pixel-level ZSAD improvement of $\texttt{FAPrompt}$ over AnomalyCLIP is marginal on part of the industrial datasets due to relatively small sample size and/or simpler pixel-level anomaly patterns in these datasets, but the improvement is large on the other industrial datasets that have a larger sample size or more complex anomaly patterns like DAGM. The latter case is particularly true on the medical image datasets, where the pixel-level anomaly patterns are more difficult to detect under the ZSAD setting, leading to AUROC performance typically below 0.90. On such challenging datasets like DAGM and medical image datasets, $\texttt{FAPrompt}$ achieves large pixel-level ZSAD performance over all competing methods, including the best competing method AnomalyCLIP.
>
> Moreover, we'd like to highlight the significant image-level ZSAD improvement of $\texttt{FAPrompt}$ over all competing methods across both industrial and medical image datasets, as shown in Table 1 in the main text.
>
> Therefore, $\texttt{FAPrompt}$ offers a substantially better ZSAD solution than existing methods in both pixel- and image-level anomaly detection.
>
> > #### **Weakness \#2. "The design of DAP is very similar to CoCoOp, and using a fixed M across images with varying scales of anomalous regions is unreasonable."**
>
> Please refer to our response to Shared Concern \#1 in the **General Response** section above for detailed clarification.
>
> > #### **Weakness \#3. "Missing necessary baseline: AnomalyCLIP with an ensemble of multiple abnormality prompts with orthogonal constraint loss; otherwise, it is difficult to justify the fine-grained prompts."**
>
> Thank you for your insightful feedback regarding the baselines. Please refer to our response to Shared Concern \#2 in the **General Response** section above for detailed clarification and empirical evidence of the significance of compound design in fine-grained prompts.

---

> ### Author Response · Authors · 2024-11-22
> **Responses to Reviewer TTqm**
>
> > #### **Questions \#1. "Could you provide additional evidence to substantiate the claim that the prompt-wise anomaly scores visualized in Figure 3 demonstrate the discriminability of $\texttt{FAPrompt}$? A comparison with visualizations from baseline models would effectively highlight the advantages."**
>
> Thanks for the valuable suggestion. Please refer to our response to Shared Concern \#3 in the **General Response** section above for clarification and empirical evidence on the ability of learning fine-grained abnormalities.
>
> > #### **Questions \#2. "To better verify the necessity of using multiple prompts, it is suggested to include more detailed ablation experiments, such as k=1 with and without CAP/DAP, similar to those in Table 4."**
>
> Please refer to our response to Shared Concern \#4 in the **General Response** section above for detailed discussion.
>
> > #### **Questions \#3. "Ablation studies on the number of layers with learnable tokens and the length of the tokens should be included."**
>
> We would like to clarify that the learnable token design follows the settings used in AnomalyCLIP and is not part of our main contributions. We use exactly the same design in this part to ensure that the performance gain/loss of $\texttt{FAPrompt}$ compared to AnomalyCLIP is not affected by this part. For clarity, a detailed description of the learnable tokens is included in the original/revised manuscript ($\texttt{Appendix}$ B.1).
>
> To evaluate the sensitivity, we also conduct additional ablation studies on the learnable tokens. The results are shown in the tables below:
>
> **# Layers having learnable tokens：**
>
> | Model  | Medical Datasets |                      | Industrial Datasets |                  |
> | :----: | :-----------------: | :------------------: | :--------------: | :--------------: |
> |        |   **Image-level**   |   **Pixel-level**    | **Image-level**  | **Pixel-level**  |
> | **5**  |  (**91.2**, 93.0)   |     (84.6, 65.0)     | (88.0, **87.3**) | (94.2, **85.5**) |
> | **7**  |  (91.0, **93.3**)   |     (85.3, 65.2)     |   (88.0, 86.9)   |   (94.6, 84.3)   |
> | **9**  |    (90.9, 93.0)     | (**85.4**, **65.9**) | (**88.2**, 87.2) | (**95.0**, 85.0) |
> | **11** |    (90.5, 92.7)     |     (84.5, 63.5)     |   (88.1, 87.2)   |   (94.9, 84.5)   |
>
> **# Length of learnable token：**
>
> | Model | Medical Datasets  |                      |   Industrial Datasets   |                  |
> | :---: | :------------------: | :------------------: | :------------------: | :--------------: |
> |       |   **Image-level**    |   **Pixel-level**    |   **Image-level**    | **Pixel-level**  |
> | **2** |     (90.7, 91.7)     |     (84.9, 65.1)     |     (88.4, 87.4)     |   (95.0, 84.8)   |
> | **4** |     (90.9, 93.0)     | (**85.4**, **65.9**) |     (88.2, 87.2)     | (**95.0**, 85.0) |
> | **6** | (**91.2**, **93.5**) |     (85.0, 65.2)     | (**90.0**, **87.7**) | (94.8, **85.3**) |
> | **8** |     (90.6, 92.3)     |     (85.0, 65.1)     |     (87.8, 86.6)     |   (94.9, 84.3)   |
>
> As shown, the performance generally gets improved with an increasing number of layers, reaching optimal performance at 9 layers. Beyond 9 layers, it tends to over-generalization, leading to a decrease in the detection performance.  A similar pattern was observed with the token length, where $\texttt{FAPrompt}$  achieves the best overall performance with a token length of 4 and 6. We have included related results in $\texttt{Appendix}$ C.5 of the revised manuscript.
>
>
>
> > #### **Questions \#4. "Minor: I recommend merging Table 3 and Table 4 for improved method comparison."**
>
> Thank you for the suggestion. We have combined Table 3 and Table 4 into a single table in the revised manuscript.

---

> > ### Author Response · Authors · 2024-12-02
> >
> > Dear Reviewer TTqm,
> >
> > The author-reviewer discussion will end soon. Please kindly check whether our response helps address your concerns or not. We're more than happy to provide more details or empirical justification if you have any follow-up questions. Thank you!
> >
> > Best regards,
> >
> > Authors of Paper 9942

---

> > > ### Comment · Reviewer_TTqm · 2024-12-03
> > >
> > > Thanks the authors for the detailed answers. However, I tend to maintain my rating after reading the responses along with the comments of the other reviewers, due to insufficient novelty and marginal improvement.

---

> > > > ### Author Response · Authors · 2024-12-04
> > > > **Responses to Reviewer TTqm**
> > > >
> > > > Thank you very much for your follow-up comment, but we would like to kindly point out that your assessment of insufficient novelty and marginal improvement is incorrect. We address this issue with the key points below.
> > > >
> > > > **Major Novelty.** At the framework level, we are the first work for learning fine-grained abnormality prompts without fine-grained text description/annotation. At the specific module level, both modules in $\texttt{FAPrompt}$ are innovative. The two modules include the compound abnormality prompting, *i.e.*, the CAP module (as elaborated in our response to Reviewer agpf Weakness #1) and the abnormality prior learning, *i.e.*, the DAP module (as elaborated in our response to Shared Concern #1), both of which represent novel contributions that have not been explored in previous works on anomaly detection or other related areas. These innovations are specifically designed to address challenges in zero-shot anomaly detection, providing new insights and solutions that well distinguish our work from existing methods.
> > > >
> > > > **Significant Performance Improvement.** The ZSAD inference process does not involve fine-tuning or adaptation to target datasets, making it highly challenging, particularly when dealing with data from unseen domains. Despite these constraints, $\texttt{FAPrompt}$ demonstrates significant improvements over the best competing method, AnomalyCLIP, in both image-level and pixel-level ZSAD performance across industrial and medical datasets. The averaged improvement is summarized in the table below (with breakdown results available in the main results section of the manuscript). We believe that it is NOT fair to look at only the results on a few selected datasets where $\texttt{FAPrompt}$ performs only slightly better than the competing methods in pixel-level AD metrics, while neglecting the significant improvement gained by $\texttt{FAPrompt}$ on the rest of the datasets and the significant improvement across the datasets on image-level AD as well. This unfairness is very undesired when numerous efforts were put to obtain the results on a large collection of datasets, including 13 image-level AD datasets and 14 pixel-level AD datasets. We think you might agree with us that we should not encourage such a paper evaluation approach in the community.
> > > >
> > > > |        Model        | Industrial Datasets  |                      |   Medical Datasets   |                      |
> > > > | :-----------------: | :------------------: | :------------------: | :------------------: | :------------------: |
> > > > |                     |   **Image-level**    |   **Pixel-level**    |   **Image-level**    |   **Pixel-level**    |
> > > > |   **AnomalyCLIP**   |     (85.0, 83.6)     |     (94.4, 84.8)     |     (87.7, 90.6)     |     (83.2, 62.9)     |
> > > > | $\texttt{FAPrompt}$ | (**88.2**, **87.2**) | (**95.0**, **85.0**) | (**90.9**, **93.0**) | (**85.4**, **65.9**) |
> > > >
> > > > We hope this explanation underscores the novelty and impact of our work. Thank you once again for your valuable evaluation and feedback.

---

### Author Response · Authors · 2024-11-22
**General Response (1/5)**

We sincerely thank all the reviewers for their time, insightful feedback, and constructive suggestions. We have carefully addressed the concerns that were common across multiple reviewers below, followed by detailed responses to individual questions and comments from each reviewer. Changes in the manuscript (beyond typos) are highlighted in **orange** in the revised version of the paper. We hope they help address the reviewers’ concerns and we are looking forward to engaging in an interesting discussion.

> ### **Shared Concern \#1. Difference between DAP and CoCoOP, and the fixed Value of $M$ [TTqm, agpf, RtD8].**

**Conceptual difference between DAP and CoCoOP.**  While the data-dependent adaptation in DAP is inspired by CoCoOp, as we mentioned in Line 261-262, there are significant differences in terms of both intuition and  design:
- **Intuition.**  DAP is designed in a way to adaptively capture abnormality-aware local features in each image as the abnormality prior. This allows our method to concentrate on various localized abnormal features for learning fine-grained abnormality in the other module of $\texttt{FAPrompt}$. In contrast, CoCoOp focuses on global class semantic in the images, which involves **no** i) adaptive local normality/abnormality modeling and ii) interaction with fine-grained pattern learning modules, rendering it ineffective for our anomaly detection objectives.
 - **Design.**  To ensure that DAP can adaptively and accurately capture the sample-wise local abnormalities, we introduce a novel abnormality prior learning loss that is specifically designed for our task and not involved in any way in CoCoOp. This loss function mitigates the influence of normal samples on abnormality tokens, allowing the learning of abnormality prompt to concentrate on truly abnormal features from abnormal samples.

These two unique characteristics are key innovations of DAP, well differentiating it from CoCoOp.

**Difference in detection performance.**  In addition, the comparative results between CoCoOp and $\texttt{FAPrompt}$ demonstrate that these designs effectively enhance both image- and pixel-level ZSAD performance (See Table 1 and Table 2). Even in our ablation study, when we removed the CAP module, the performance of DAP alone still surpasses CoCoOp (See Table 3). This further demonstrates the difference of our DAP module and its significance to ZSAD.

**Hyperparameter $M$.**  Regarding the fixed value of $M$ across images, we have conducted a sensitivity analysis in the original manuscript, as illustrated in Figure 4. The abnormality prior in DAP is derived from the top $M$ most abnormal patches. Setting $M$ too large can introduce noise or irrelevant patches, diminishing the abnormality prompt’s effectiveness. In contrast, a very small $M$ might fail to capture sufficient local abnormality features. Our experiments indicate that $M = 10$ is a balanced choice, resulting in superior performance across the datasets.

---

> ### Author Response · Authors · 2024-11-23
> **General Response (2/5)**
>
> > ### **Shared Concern \#2. Significance of Compound design in Abnormality Prompts [TTqm, nnbo, agpf].**
>
> We conduct additional experiments for AnomalyCLIP with an ensemble of multiple abnormality prompts and an orthogonal constraint loss, denoted by AnomalyCLIP Ensemble*. The results are shown in the table below, which is included in Table 3 in the revised manuscript:
> |           Model           | Industrial Datasets  |                      |   Medical Datasets   |                      |
> | :-----------------------: | :------------------: | :------------------: | :------------------: | :------------------: |
> |                           |   **Image-level**    |   **Pixel-level**    |   **Image-level**    |   **Pixel-level**    |
> |      **AnomalyCLIP**      |     (85.0, 83.6)     |     (94.4, 84.8)     |     (87.7, 90.6)     |     (83.2, 62.9)     |
> | **AnomalyCLIP Ensemble**  |     (85.5, 84.0)     |   (94.7, **85.0**)   |     (89.3, 91.3)     |     (83.2, 62.4)     |
> | **AnomalyCLIP Ensemble*** |     (85.5, 82.6)     |     (94.6, 84.5)     |     (88.8, 91.0)     |     (83.5, 65.6)     |
> |    $\texttt{FAPrompt}$    | (**88.2**, **87.2**) | (**95.0**, **85.0**) | (**90.9**, **93.0**) | (**85.4**, **65.9**) |
>
> As shown, AnomalyCLIP Ensemble* shows only some modest improvement over the baseline AnomalyCLIP, and it is not always more effective than the simple AnomalyCLIP ensemble approach,  AnomalyCLIP Ensemble (already included in original manuscript).
>
> However, $\texttt{FAPrompt}$ consistently outperforms this ensemble method across both image-level and pixel-level ZSAD results. This better performance of $\texttt{FAPrompt}$ can be attributed to the effectiveness of compounded prompt design in multiple abnormality prompts and abnormality prior from the DAP module. This helps justify that the fine-grained abnormalities learned in $\texttt{FAPrompt}$ cannot be learned in simple prompt ensemble approaches.

---

> ### Author Response · Authors · 2024-11-23
> **General Response (3/5)**
>
> > ### **Shared Concern \#3. Additional Evidence for Verifying Significance of Fine-grained Abnormality Prompts in Figure 3 [TTqm, nnbo, agpf].**
>
> We provide t-SNE visualization and quantitative results of AnomalyCLIP  and the prompt ensemble method AnomalyCLIP Ensemble* for their comparison with $\texttt{FAPrompt}$ on the datasets. The results are included in Figure 6, page 20 of the revised manuscript ($\texttt{Appendix}$ C.3). Note that the difference between AnomalyCLIP and $\texttt{FAPrompt}$/AnomalyCLIP Ensemble*  in the figure is because AnomalyCLIP learns one single abnormality prompt only while the $\texttt{FAPrompt}$/AnomalyCLIP Ensemble* learns 10 abnormality prompts.
>
> - **$\texttt{FAPrompt}$ vs. AnomalyCLIP.**  It is clear that compared to AnomalyCLIP, $\texttt{FAPrompt}$ learns a set of effective complementary abnormal patterns captured by the 10 abnormality prompts, resulting in better detection performance on datasets with complex anomaly cases. For example, on the datasets BTAD(01) and VisA (pcb4), several anomalies distributed very closely to, or overlapped with part of the normal images, are difficult to detect using single abnormality prompt in AnomalyCLIP, indicating that this abnormality prompt is not discriminative w.r.t. these anomalies. In contrast, $\texttt{FAPrompt}$ alleviates this situation with the abnormality prompts that show visually different, discriminative power.
>
>   For datasets with simpler patterns like VisA (chewinggum), single abnormality prompt is sufficient, while having multiple abnormality prompts in $\texttt{FAPrompt}$ do not have adverse effects. This demonstrates the capability of $\texttt{FAPrompt}$ in achieving stable, effective detection across simple and complex datasets.
>
> - $\texttt{FAPrompt}$ **vs. the prompt ensemble method AnomalyCLIP Ensemble$*$.**  Despite also learning multiple abnormality prompts, it is clear from the visualization that the abnormality prompts in AnomalyCLIP Ensemble* tend to be clustered closely, while that in $\texttt{FAPrompt}$ is much more disperse, *e.g.*, two clustered patterns on BTAD(01) and one clustered pattern on VisA (pcb4) learned by AnomalyCLIP Ensemble* vs. four disperse patterns on both datasets learned by $\texttt{FAPrompt}$. Importantly, the more disperse abnormal patterns from $\texttt{FAPrompt}$ provides complementary discriminative power to each other, substantiated by the enhanced AUROC/AP performance compared to AnomalyCLIP Ensemble*.

---

> ### Author Response · Authors · 2024-11-23
> **General Response (4/5)**
>
> > ### **Shared Concern \#4. Additional Results for $\texttt{FAPrompt}$ with $K=1$ [TTqm, nnbo, RtD8].**
>
> We conduct additional experiments for $\texttt{FAPrompt}$ with $K = 1$.  The results are shown in table below:
> |          Model           | Industrial Datasets  |                      | Medical Datasets |                      |
> | :----------------------: | :------------------: | :------------------: | :--------------: | :------------------: |
> |                          |   **Image-level**    |   **Pixel-level**    | **Image-level**  |   **Pixel-level**    |
> |     **AnomalyCLIP**      |     (85.0, 83.6)     |     (94.4, 84.8)     |   (87.7, 90.6)   |     (83.2, 62.9)     |
> |        **+ DAP**         |     (86.9, 85.2)     |     (94.8, 84.9)     |   (90.2, 92.3)   |     (84.6, 64.8)     |
> |    **+ CAP ($K=1$)**     |     (85.7, 85.5)     |     (94.5, 83.9)     |   (89.9, 91.3)   |     (83.6, 63.8)     |
> | **+ CAP ($K=1$) + DAP**  |     (86.1, 86.2)     |     (94.3, 83.7)     |   (90.4, 91.3)   |     (83.9, 63.5)     |
> |    **+ CAP ($K=10$)**    |     (88.1, 87.0)     |     (94.6, 83.9)     | (90.6, **93.1**) |     (83.8, 63.8)     |
> | **+ CAP ($K=10$) + DAP** | (**88.2**, **87.2**) | (**95.0**, **85.0**) | (**90.9**, 93.0) | (**85.4**, **65.9**) |
>
> Even without applying DAP, the $\texttt{FAPrompt}$ variant using a single compound abnormality prompt '+ CAP($K = 1)$)' also gains improved performance over the base model AnomalyCLIP. This improvement becomes more pronounced as $K$ increases to 10, which denoted as '+ CAP($K = 10$)'. This improvement indicates that multiple prompts are effective in capturing a broader spectrum of abnormalities.
>
> The combination of '+ CAP(K = 1) + DAP' underperforms compared to using DAP alone. This is because '+ CAP($K = 1$) + DAP' relies on just a single abnormality prompt with a limited set of abnormal tokens, restricting its ability to capture the full diversity of abnormalities and leverage the abnormality prior provided by DAP effectively. However, when the number of abnormality prompts increases to 10, the ability of $\texttt{FAPrompt}$ to learn diverse abnormal patterns improves substantially. As a result, '+ CAP($K = 10$) + DAP', which is also our full $\texttt{FAPrompt}$, achieves the best performance across various datasets.
>
> These results demonstrate that using multiple prompts enables $\texttt{FAPrompt}$ to better capture diverse, complementary abnormalities, maximizing the benefit of both CAP and DAP components for the overall superior performance. We have included related results in Figure 4 and Figure 7, as well as associated analysis in$ \texttt{Appendix}$ C.5 in the revised manuscript.

---

> > ### Comment · Reviewer_RtD8 · 2024-11-28
> > **Marginal gains on pixel-level results on industrial datasets**
> >
> > Thank you for providing the additional experiments. However, according to the table above, only the complete version of the proposed method (CAP (K=10) + DA) achieves slight improvements in pixel-level results for industrial datasets, while other alternatives either underperform or are comparable to the baseline AnomalyCLIP. These experimental results fail to substantiate the main idea of fine-grained anomaly learning and the effectiveness of the proposed method.

---

> > > ### Author Response · Authors · 2024-11-28
> > > **Response to Reviewer RtD8**
> > >
> > > **Complementary Learning of CAP and DAP.** We believe there might be some misunderstandings here. Both '+CAP($K=1$)' and '+CAP($K=1$) + DAP' are *naive variants* of $\texttt{FAPrompt}$ since using $K=1$ indicates that we learn only single abnormality prompt, which is opposite to the learning of multiple, complementary abnormality prompts in $\texttt{FAPrompt}$. There are devised to address the concern about whether learning $K=10$ abnormality prompts is helpful in the two modules of $\texttt{FAPrompt}$, *i.e.*, '+DAP' and '+CAP($K=10$)'. As shown by the results in the table below (copied from our response to Shared Concern \#4), '+CAP($K=10$)' and '+CAP($K=10$)+DAP' clearly outperform outperform '+CAP($K=1$)' and +'CAP($K=1$)+DAP' respectively, demonstrating the improvement of learning ten abnormality prompts over single abnormality prompt.
> > >
> > > |          Model           | Industrial Datasets  |                      | Medical Datasets |                      |
> > > | :----------------------: | :------------------: | :------------------: | :--------------: | :------------------: |
> > > |                          |   **Image-level**    |   **Pixel-level**    | **Image-level**  |   **Pixel-level**    |
> > > |     **AnomalyCLIP**      |     (85.0, 83.6)     |     (94.4, 84.8)     |   (87.7, 90.6)   |     (83.2, 62.9)     |
> > > |        **+ DAP**         |     (86.9, 85.2)     |     (94.8, 84.9)     |   (90.2, 92.3)   |     (84.6, 64.8)     |
> > > |    **+ CAP ($K=1$)**     |     (85.7, 85.5)     |     (94.5, 83.9)     |   (89.9, 91.3)   |     (83.6, 63.8)     |
> > > |    **+ CAP ($K=10$)**    |     (88.1, 87.0)     |     (94.6, 83.9)     | (90.6, **93.1**) |     (83.8, 63.8)     |
> > > | **+ CAP ($K=1$) + DAP**  |     (86.1, 86.2)     |     (94.3, 83.7)     |   (90.4, 91.3)   |     (83.9, 63.5)     |
> > > | **+ CAP ($K=10$) + DAP** | (**88.2**, **87.2**) | (**95.0**, **85.0**) | (**90.9**, 93.0) | (**85.4**, **65.9**) |
> > >
> > >
> > >
> > > As for the CAP module '+CAP(K=10)', without the abnormality prior from query/test image through the DAP module, it is less effective in learning the complementary abnormalities. As discussed in Sec.4.2 Module ablation and can also be observed in the table here, the CAP module significantly contributes to image-level ZSAD performance due to its ability in learning the fine-grained abnormality details, while the DAP module have more impact on pixel-level performance due to its ability in extracting local abnormal features as the abnormality prior. Therefore, our full $\texttt{FAPrompt}$ achieves the best results, which demonstrates the complementary abilities of the two different modules.
> > >
> > > **Substantial improvement in pixel-level ZSAD.** While pixel-level ZSAD improvements for $\texttt{FAPrompt}$ over AnomalyCLIP are marginal on simpler industrial datasets with small sample sizes or straightforward patterns, $\texttt{FAPrompt}$ achieves significant gains on more complex datasets, such as DAGM and medical datasets, as shown in Table 2 in the paper. These datasets are characterized by intricate pixel-level anomaly patterns and AUROC values typically below 0.90 under ZSAD settings. Large improvement on them demonstrates the importance of learning of fine-grained complementary abnormalities to achieve more promising pixel-level performance on complex datasets.
> > >
> > > **Significant improvement in image-level ZSAD.** Moreover, we'd like to highlight the significant image-level ZSAD improvement of $\texttt{FAPrompt}$ over all competing methods across both industrial and medical image datasets, as shown in Table 1 in the main text.
> > >
> > > Overall, $\texttt{FAPrompt}$ offers a substantially better ZSAD solution than existing methods in both pixel- and image-level anomaly detection. The improvement on image-level ZSAD is consistently significant on both industrial and medical datasets. While the improvement on pixel-level ZSAD can be marginal on industrial datasets, it is significant on the medical datasets, i.e., $\texttt{FAPrompt}$ (85.4) vs AnomalyCLIP (83.2) in AUROC and $\texttt{FAPrompt}$ (65.9) vs AnomalyCLIP (62.9) in PRO. These results justifies the effectiveness of improving the single abnormality learning in AnomalyCLIP with multiple fine-grained abnormality prompt learning.

---

> ### Author Response · Authors · 2024-11-23
> **General Response (5/5)**
>
> > ### **Shared Concern \#5. Significance to Averaging Strategy in CAP [nnbo, RtD8].**
>
> Despite the simplicity, the use of the averaging operation is due to its general effectiveness in aggregating multiple patterns. This strategy is also widely used in existing ZSAD and FSAD methods, such as WinCLIP and AnoVL, to deal with diverse and complementary abnormality text information.
>
> To validate its advantage over the alternatives, we conduct additional experiments to evaluate two variants of $\texttt{FAPrompt}$, with the results presented in table below:
>
> |           Model           | Industrial Datasets  |                      |   Medical Datasets   |                      |
> | :-----------------------: | :------------------: | :------------------: | :------------------: | :------------------: |
> |                           |   **Image-level**    |   **Pixel-level**    |   **Image-level**    |   **Pixel-level**    |
> |      **AnomalyCLIP**      |     (85.0, 83.6)     |     (94.4, 84.8)     |     (87.7, 90.6)     |     (83.2, 62.9)     |
> | $\texttt{FAPrompt}_{0.1}$ |     (87.9, 87.0)     |     (93.0, 82.2)     |     (90.6, **93.0**)     |     (84.4, 65.1)     |
> | $\texttt{FAPrompt}_{0.2}$ |     (87.7, 86.7)     |     (94.4, 83.2)     |   (**90.9**, 92.3)   |     (85.1, 65.7)     |
> |    $\texttt{FAPrompt}$    | (**88.2**, **87.2**) | (**95.0**, **85.0**) | (**90.9**, **93.0**) | (**85.4**, **65.9**) |
>
> **$\texttt{FAPrompt}_{0.1}:$ Selecting the Most Similar Prompt for Each Detected Abnormality.**
>
> In this variant of $\texttt{FAPrompt}$, we calculate the cosine similarity between the individual abnormality prompts and each test image to select the similarity to the most similar prompt as the anomaly score during inference. While this approach shows comparable performance on image-level ZSAD results, it can largely underperform the primary $\texttt{FAPrompt}$ in pixel-level ZSAD. This is mainly because selecting only a single prompt can lead to the loss of complementary information from other abnormality prompts, limiting the model's ability to detect the full spectrum of abnormalities.
>
> **$\texttt{FAPrompt}_{0.2}:$ Using Weighted Abnormality Prompts.**
>
> In this variant, we use a prompt importance learning network to learn a set of weights for each abnormality prompt based on the selected most abnormal patch tokens of the query images. These weights are then used to combine multiple abnormality prompts into a single weighted abnormality prompt (a weighted abnormality prototype) for ZSAD.
>
> Although this variant outperforms  $\texttt{FAPrompt}_{0.1}$ by retaining some complementary abnormality information, it does not match the performance of the simple averaging. This may be due to the greater power of the model in fitting the query images, which can lead to overfitting of the tuning auxiliary dataset in the zero-shot setting, *i.e.*, the learned weights may well reflect the significance of each prompt in the tuning dataset but not in the target datasets.
>
> Given these results, we chose to average the abnormality prompts in $\texttt{FAPrompt}$, as it offers a straightforward yet effective way to integrate diverse abnormalities while preserving their complementary information. We have included related results and analysis in $\texttt{Appendix}$ C.4 in the revised manuscript.

---

> > ### Comment · Reviewer_RtD8 · 2024-11-26
> > **Doubts on averaging strategy in CAP.**
> >
> > I appreciate the authors' additional experiments exploring alternative methods of leveraging abnormality prompts. The results of $\rm FAPrompt_{0,1}$ demonstrate the effectiveness of using a single best-matched abnormality prompt for image-level anomaly detection. However, the design of $\rm FAPrompt_{0,2}$ is less convincing. Simply averaging or weighting abnormality prompts still results in a single text embedding, whereas each image patch is expected to correspond to its own best-matched abnormality prompt, aligning with the concept of fine-grained abnormality prompts.

---

> ### Author Response · Authors · 2024-11-27
> **Response to Reviewer RtD8**
>
> Many thanks for your follow-up suggestion.
> We have conducted additional experiments with another alternative variant of  $\texttt{FAPrompt}$, $\texttt{FAPrompt}_{0.3}$.
>
> In this variant, we use the cosine similarity of each image patch token to its most similar abnormality prompt to obtain the pixel-level anomaly scores and generate the pixel-level anomaly detection map. In the same way as $\texttt{FAPrompt}$, the image-level anomaly score is then derived from the cosine similarity of the image token to its most similar abnormality prompt, combined with the maximum score from the pixel-level anomaly detection map. The image- and pixel-level AD results of $\texttt{FAPrompt}_{0.3}$ are reported in the table below.
>
> |           Model           | Industrial Datasets  |                      |   Medical Datasets   |                      |
> | :-----------------------: | :------------------: | :------------------: | :------------------: | :------------------: |
> |                           |   **Image-level**    |   **Pixel-level**    |   **Image-level**    |   **Pixel-level**    |
> |      **AnomalyCLIP**      |     (85.0, 83.6)     |     (94.4, 84.8)     |     (87.7, 90.6)     |     (83.2, 62.9)     |
> | $\texttt{FAPrompt}_{0.1}$ |     (87.9, 87.0)     |     (93.0, 82.2)     |   (90.6, **93.0**)   |     (84.4, 65.1)     |
> | $\texttt{FAPrompt}_{0.2}$ |     (87.7, 86.7)     |     (94.4, 83.2)     |   (**90.9**, 92.3)   |     (85.1, 65.7)     |
> | $\texttt{FAPrompt}_{0.3}$ |     (85.1, 84.9)     |   (**95.0**, 83.2)   |     (85.6, 89.1)     |   (84.7, **65.9**)   |
> |    $\texttt{FAPrompt}$    | (**88.2**, **87.2**) | (**95.0**, **85.0**) | (**90.9**, **93.0**) | (**85.4**, **65.9**) |
>
> The results indicate that while $\texttt{FAPrompt}_{0.3}$ achieves comparably good performance in pixel-level AD, it substantially underperforms the simply averaging method ($\texttt{FAPrompt}$) in image-level AD. This further supports our observation that relying on a single prompt can result in the loss of complementary information provided by the other abnormality prompts, thereby limiting the model's ability to effectively capture the full spectrum of abnormalities; and averaging the abnormality prompts serves as a simple yet effective way to aggregate the different abnormalities from the multiple abnormality prompts learned by $\texttt{FAPrompt}$.
>
> Additionally, the strategy used in $\texttt{FAPrompt}_{0.3}$ can incur significant computational overhead, since it requires repeated computation to select the most similar prompt for each  patch tokens, i.e., given an image having 1,369 (37x37) patch tokens in our experiments, there would be a computational overhead of $(1,370 \times 10 - 1,370 \times 1)$ similarity calculation per test image.
>
> Therefore, in terms of either effectiveness or efficiency, averaging is a straightforward but more cost-effective way to integrate diverse abnormalities while preserving their complementary information.

---

> > ### Comment · Reviewer_RtD8 · 2024-11-28
> > **Doubts on image-level comparison.**
> >
> > Thank you for the authors' follow-up response. Regarding $\rm FAPrompt_{0,3}$, how are the learnable abnormality prompts obtained? Are these abnormality prompts still averaged during training? If so, the learned individual abnormality prompts may not align with the image patch tokens during testing due to inconsistencies in the use of abnormality prompts between the training and testing phases.
> >
> > Additionally, the image-level results are less convincing, as matching a patch with an individual prompt affects two variables: the cosine similarity between the image token and its most similar abnormality prompt, and the pixel-level anomaly detection map. Moreover, the optimal image-level anomaly scores are not necessarily the equal-weighted average of these two terms in the case of $\rm FAPrompt_{0,3}$.
> >
> > For a fair comparison with $\rm FAPrompt$  in image-level detection, would it be possible to rely solely on either the cosine similarity of the image token to its most similar abnormality prompt or the maximum score from the pixel-level anomaly detection map?

---

> > > ### Author Response · Authors · 2024-11-28
> > > **Response to Reviewer RtD8**
> > >
> > > Thank you very much for the feedback and questions.
> > > The learnable abnormality prompts in $\texttt{FAPrompt}_{0.3}$ are not averaged during either training or testing. Instead, the pixel-level anomaly scores are calculated based on the similarity between patch token embeddings and their best-match abnormality prompt. For the image-level results, we calculate the similarity between the image token embedding and its best-match abnormality prompt.
> > >
> > > To address your further concern on image-level results, we derive another variant $\texttt{FAPrompt}_{0.4}$.
> > >
> > > This variant is the same as $\texttt{FAPrompt}_{0.3}$ except that we calculate the image-level anomaly scores using only the maximum value of the patch-wise anomaly score map.
> > >
> > > As shown in the table below, while this variant improves on the image-level ZSAD performance over $\texttt{FAPrompt}_{0.3}$, it still underperforms the original $\texttt{FAPrompt}$ by an AUROC/AP of 2%. We can safely attribute the improved performance of $\texttt{FAPrompt}$ over these two variants to the use of multiple complementary abnormality prompts, since that is the only difference between $\texttt{FAPrompt}$ and them.
> > >
> > > |           Model           | Industrial Datasets  |                      |   Medical Datasets   |                      |
> > > | :-----------------------: | :------------------: | :------------------: | :------------------: | :------------------: |
> > > |                           |   **Image-level**    |   **Pixel-level**    |   **Image-level**    |   **Pixel-level**    |
> > > |      **AnomalyCLIP**      |     (85.0, 83.6)     |     (94.4, 84.8)     |     (87.7, 90.6)     |     (83.2, 62.9)     |
> > > | $\texttt{FAPrompt}_{0.3}$ |     (85.1, 84.9)     |   (**95.0**, 83.2)   |     (85.6, 89.1)     |   (84.7, **65.9**)   |
> > > | $\texttt{FAPrompt}_{0.4}$ |     (86.9, 86.6)     |   (**95.0**, 83.2)   |     (88.9, 90.9)     |   (84.7, **65.9**)   |
> > > |    $\texttt{FAPrompt}$    | (**88.2**, **87.2**) | (**95.0**, **85.0**) | (**90.9**, **93.0**) | (**85.4**, **65.9**) |

---

> > > > ### Comment · Reviewer_RtD8 · 2024-11-29
> > > > **Unfair comparison with $\texttt{FAPrompt}$**
> > > >
> > > > Thanks for providing more experiments and sorry for the confusion caused by my comments. In image-level comparison, $\rm {FAPrompt}$ appears to derive the resulting image-level anomaly score by averaging the image-text similarity score and the maximum pixel-level anomaly score, i.e., Eq. (9). It would be fairer to use only the maximum value of the patch-wise anomaly score map to calculate the image-level anomaly score for $\texttt{FAPrompt}$.

---

> > > > > ### Author Response · Authors · 2024-11-30
> > > > > **Response to Reviewer RtD8 (3/3)**
> > > > >
> > > > > **Complexity Comparison.** Moreover, as mentioned in our previous response, the strategy of selecting the best-matched individual abnormality prompt in $^{\ast}\texttt{FAPrompt}$ can incur significant computational overhead. This is because it requires repeated computations to identify the most similar prompt for each patch token embedding. For instance, in our experiments, an image with 1,369 ($37\times37$) patch tokens leads to substantial overhead for each test image. However, this additional computational cost does not result in any performance improvements.
> > > > >
> > > > > Therefore, while alternative strategies to synthesize the multiple complementary prompts may be further explored in future work, we believe that the simple averaging is an effective strategy within our framework, and it indeed delivers the best performance in learning the complementary abnormalities compared to its best-matched prompting methods and existing state-of-the-art competing methods.
> > > > >
> > > > > We sincerely hope that this comprehensive response fully addresses your concern regarding the averaging strategy. Thank you very much again for the insightful feedback.
> > > > >
> > > > > **Reference**
> > > > >
> > > > > [1] Jongheon Jeong, Yang Zou, Taewan Kim, Dongqing Zhang, Avinash Ravichandran, and Onkar
> > > > > Dabeer. Winclip: Zero-/few-shot anomaly classification and segmentation. In Proceedings of the IEEE/CVF Conference on Computer Vision and Pattern Recognition, 2023.
> > > > >
> > > > > [2] Qihang Zhou, Guansong Pang, Yu Tian, Shibo He, and Jiming Chen. Anomalyclip: Object-agnostic prompt learning for zero-shot anomaly detection. In The Twelfth International Conference on Learning Representations, 2024.
> > > > >
> > > > > [3] Chen, Xuhai and Han, Yue and Zhang, Jiangning. APRIL-GAN: A Zero-/Few-Shot Anomaly Classification and Segmentation Method for CVPR 2023 VAND Workshop Challenge Tracks 1\&2: 1st Place on Zero-shot AD and 4th Place on Few-shot AD.
> > > > >
> > > > > [4] Jiawen Zhu and Guansong Pang. Toward generalist anomaly detection via in-context residual
> > > > > learning with few-shot sample prompts. In Proceedings of the IEEE/CVF Conference on
> > > > > Computer Vision and Pattern Recognition, 2024.
> > > > >
> > > > > [5] Li, Xiaofan and Zhang, Zhizhong and Tan, Xin and Chen, Chengwei and Qu, Yanyun and Xie, Yuan and Ma, Lizhuang. PromptAD: Learning Prompts with only Normal Samples for Few-Shot Anomaly Detection. In The Twelfth International Conference on Learning Representations, 2024.

---

> > > > > ### Author Response · Authors · 2024-12-02
> > > > >
> > > > > Dear Reviewer **RtD8**,
> > > > >
> > > > > Please be kindly advised that we have provided a detailed response to address your concern on the use of fine-grained abnormality prompts learned by our method. We would greatly appreciate if you could go through it and re-consider your rating. Thank you very much!
> > > > >
> > > > > Best regards,
> > > > >
> > > > > Authors of Paper 9942

---

> ### Author Response · Authors · 2024-11-30
> **Response to Reviewer RtD8 (1/3)**
>
> Thank you very much for your comment. Since there are various ways to calculate the anomaly scores during training and inference of $\texttt{FAPrompt}$, to avoid confusion, we re-organize the results of $\texttt{FAPrompt}$, together with some additional results, to address your concern on how the fine-grained abnormality prompts should be used.
>
> **Notation.** We use the below notations for better clarity.
>
> - **Training.**  We use $^{\ast}$$\texttt{FAPrompt}$ and $\texttt{FAPrompt}$ to represent two different ways of using the abnormality prompts during training. In particular, $^{\ast}$$\texttt{FAPrompt}$ denotes the variant of $\texttt{FAPrompt}$ that uses the cosine similarity of each patch token to its most similar abnormality prompt to obtain the pixel-level anomaly scores and generate the pixel-level anomaly detection map. Similarly, $^{\ast}$$\texttt{FAPrompt}$ uses the similarity between the image embedding and its most similar abnormality prompt to define the image-level similarity score, *i.e.*, $s_a(x)$. $\texttt{FAPrompt}$ represents the default $\texttt{FAPrompt}$ in our paper that uses the abnormality prompt prototype generated from the $K$ multiple complementary abnormality prompts in the above two similarity calculations, rather than the best-matched individual abnormality prompt.
> - **Testing.**  At the testing stage, both models have three ways to calculate the image-level anomaly scores, including the use of the maximum **p**ixel-level anomaly score solely, *i.e.*, $s^{\prime}_a(x)$ in our paper (we denote it by NAME$-\text{P}$ here),  the use of **i**mage-level similarity score solely, *i.e.*, $s_a(x)$ in our paper (denoted by NAME$-\text{I}$), and their averaged combination as defined in Eq. 9 in our paper (denoted by NAME$-\text{PI}$), where NAME can be replaced by either $\texttt{FAPrompt}$ or $^{\ast}$$\texttt{FAPrompt}$. Please be noted that the similarity calculation in both $^{\ast}$$\texttt{FAPrompt}$ and $\texttt{FAPrompt}$ during testing is consistent with those used in the training, *i.e.*, the best-matched abnormality prompt is used in $^{\ast}$$\texttt{FAPrompt}$ and the abnormality prompt prototype is used in $\texttt{FAPrompt}$.
>
> If we'd like to align these new notations with the ones used in the previous responses, $^{\ast}\texttt{FAPrompt}-\text{PI}$ is $\texttt{FAPrompt}_{0.3}$.
>
> $\texttt{FAPrompt}-\text{PI}$ is the default $\texttt{FAPrompt}$ model used in the experiments in our paper. If we understand correctly, you requested the results of $\texttt{FAPrompt}-\text{P}$ in the latest comment.

---

> ### Author Response · Authors · 2024-11-30
> **Response to Reviewer RtD8 (2/3)**
>
> **Comparison of Image-level Results.** The image-level results (AUROC, AP) are presented in the table below. We analyze each of the three image-level anomaly scoring methods for both $^{\ast}\texttt{FAPrompt}$ and $\texttt{FAPrompt}$ in detail in the following.
>
> |                      Model                      | s_a(x)' | s_a(x) | Image-level Results |              |
> | :---------------------------------------------: | :-----: | :----: | :-----------------: | :----------: |
> |                                                 |         |        |     Industrial      |   Medical    |
> | **Training and Testing on Best-matched Prompt** |         |        |                     |              |
> |       $^{\ast}\texttt{FAPrompt}-\text{P}$       |    √    |   ×    |    (86.9, 86.6)     | (88.9, 90.9) |
> |       $^{\ast}\texttt{FAPrompt}-\text{I}$       |    ×    |   √    |    (75.6, 72.7)     | (67.5, 75.0) |
> |      $^{\ast}\texttt{FAPrompt}-\text{PI}$       |    √    |   √    |    (85.1, 84.9)     | (85.6, 89.1) |
> | **Training and Testing on the Average Prompt**  |         |        |                     |              |
> |          $\texttt{FAPrompt}-\text{P}$           |    √    |   ×    |    (87.2, 86.6)     | (88.6, 90.1) |
> |          $\texttt{FAPrompt}-\text{I}$           |    ×    |   √    |    (85.6, 84.0)     | (90.4, 92.2) |
> |          $\texttt{FAPrompt}-\text{PI}$          |    √    |   √    |    (**88.2**, **87.2**)     | (**90.9**, **93.0**) |
>
> - **Using maximum pixel-level anomaly score as image-level anomaly score ($s_a^\prime(x)$)**
>
>   Both $^{\ast}\texttt{FAPrompt}-\text{P}$ and $\texttt{FAPrompt}-\text{P}$ achieve strong image-level ZSAD performance in this case because the maximum pixel-level anomaly score effectively captures local abnormality information, which can be used to distinguish between normal and abnormal samples. This indicates that both the best-matched abnormality prompt in $^{\ast}\texttt{FAPrompt}-\text{P}$ and the averaged abnormality prompt in $\texttt{FAPrompt}-\text{P}$ are highly effective in representing pixel-level information within the patch token embeddings.
>
> - **Using image-level similarity score as image-level anomaly score ($s_a(x)$)**
>
>   $\texttt{FAPrompt}-\text{I}$ significantly outperforms $^{\ast}\texttt{FAPrompt}-\text{I}$, with the latter yielding poor performance. This large performance gap is due to the fact that the abnormality prompt learning in $^{\ast}\texttt{FAPrompt}-\text{I}$ being strongly influenced by their alignment with the 1,369 patch token embeddings, leading to severe bias toward patch-level anomaly scores when matching these prompts to the image token embedding.
>
>   This argument is also supported by the comparably strong image-level performance of $\texttt{FAPrompt}_{0.1}$, as discussed in our previous response.
>
>   The key difference between them is that $\texttt{FAPrompt}_{0.1}$ uses the same abnormality prompt (*i.e.*, the one best-matched to the image token embedding) for both patch and image token embeddings, avoiding the bias caused by the best-matching to a large set of patch tokens, whereas $^{\ast}\texttt{FAPrompt}-\text{I}$ selects separate best-matched prompts for patch and image token embeddings.
>
>   These results highlight the stronger power of the averaged abnormality prompt in $^{\ast}\texttt{FAPrompt}-\text{I}$ in capturing global abnormality information within the image token embedding.
>
> - **Using the average of $s_a^\prime(x)$ and $s_a(x)$ as image-level anomaly score**
>
>   When using both local and global anomaly scores, $\texttt{FAPrompt}-\text{PI}$ consistently outperforms $^{\ast}\texttt{FAPrompt}-\text{PI}$ by a significant margin. This demonstrates that the averaged fine-grained abnormality prompts are more effective than the single best-matched prompt in collaboratively capturing local and glocal abnormality, as the averaging operation ensures better compatibility with both global (from image token embedding) and local (from patch token embeddings) information.
>
> We would like to kindly point out that using the average of the global (image-level $s_a(x)$) and local (patch-level $s_a(x)$) anomaly scores to define holistic anomaly scores for images is a commonly-used, effective strategy, which has been demonstrated in a number of recent zero- and few-shot anomaly detection methods (*e.g.*, WinCLIP [1], AnomalyCLIP [2], April-GAN [3], InCTRL [4], and PromptAD [5]). Our image-level anomaly scoring follows this approach and verifies its effectiveness in our $\texttt{FAPrompt}$.
>
> It is also very clear that $\texttt{FAPrompt}-\text{PI}$ is consistently the best performer compared to all the variants of $^{\ast}\texttt{FAPrompt}$. These comprehensive ablation study results well justify the effectiveness of $\texttt{FAPrompt}$ using the averaged abnormality prompt (*i.e.*, the prototype) in capturing the diverse complementary abnormalities

---

### Author Response · Authors · 2024-11-26

Dear Reviewers,

Thank you once again for your constructive feedback and thoughtful suggestions. We would like to kindly remind you that we have thoroughly addressed your concerns by clarifying our contributions, emphasizing the significant performance improvements of our method, and including additional comparisons with baseline models as well as ablation studies.

As the paper revision period is near to close, we look forward to hearing your response and remain available to address any further comments or questions you might have.

Best regards,
Authors

---

### Meta-Review · Area_Chair_Z2fp · 2024-12-16

**Metareview:**

The paper is well-written and the experimental results are strong and the figures are well-drawn. What reviewers are most concerned about is the novelty of this paper, they recognized that the ideas if this paper come from the prompt tuning and prototype learning, like CoCoOp. The author's careful response solves part of the problems. However, AC still believes that the inspiration and insights of this paper is not novel enough to meet the standards of this top meeting. So the final vote is rejection.

**Additional Comments On Reviewer Discussion:**

The contribution of this paper is kinda different from the existing work. But all reviewers do not agree that this paper is very innovative.

---

### Decision · Program_Chairs · 2025-01-22

Reject